

# Detection of landslide timing, reactivation and precursory motion during the 2018, Lombok, Indonesia earthquake sequence with Sentinel-1

Katy Burrows[1,2], David, G. Milledge[3], and Maria Francesca Ferrario[4]

[1]Università degli Studi di Milano-Bicocca, Milan, Italy
[2]European Space Agency (ESA) ESRIN, Frascati, Italy
[3]Newcastle University, Newcastle upon Tyne, United Kingdom
[4]Università degli Studi dell'Insubria, Como, Italy

**Correspondence:** Katy Burrows (katyaline.burrows@unimib.it)

**Abstract.** Earthquake-triggered landslides can be mapped using optical satellite images, but assessing how they evolve through time during sequences of earthquakes is difficult with such data due to cloud cover. Here we use Sentinel-1 techniques to characterise the evolution of rapid landslides during the 2018 Lombok, Indonesia earthquake sequence. While the majority of new landslides were triggered during the largest earthquake in the sequence on 05/08, we are also able to identify landslide activity associated with other, lower magnitude earthquakes on 28/07, 09/08 and 19/08, with many landslides active in more than one earthquake. In particular, many landslides triggered by the 05/08 earthquake were then reactivated later in the sequence. These reactivations were triggered by accelerations as weak as 0.1g, while new failures generally did not occur below 0.15g, suggesting a post-seismic weakening effect driven by the landslides themselves rather than general landscape weakening. We also identified at least one example where precursory motion during the first earthquake in the sequence was later followed by larger scale failure. Overall, we demonstrate that Sentinel-1 amplitude and coherence are valuable tools to study how landslide hazard and mass wasting evolve during sequences of triggers.

## 1 Introduction

Earthquakes can trigger widespread landsliding, which represents a major secondary hazard and can have a significant mass wasting effect. As these landslides are often triggered across a large area, remote sensing has emerged as a vital tool to quantify earthquake-triggered landslides (Novellino et al., 2024). In particular, landslide inventories are often compiled through manual mapping by the comparison of pre- and post-seismic multi-spectral satellite images (e.g. Ferrario, 2019; Ferrario et al., 2024; Tanyaş et al., 2022; Tiwari et al., 2017) or through automated methods that use these data (Milledge et al., 2022; Scheip and Wegmann, 2021). These inventories can then be used to assess the impacts on the landscape and the potential for further hazards (Parker et al., 2011; Croissant et al., 2019), to further our understanding of the triggering process, and to build and calibrate physical and empirical models that can then be applied to future earthquakes (e.g. Godt et al., 2008; Nowicki Jessee et al., 2018).





In many cases, landslide-triggering earthquakes are accompanied by foreshocks and aftershocks that are also large enough to trigger landslides and remobilise co-seismic landslide deposits (e.g. Fan et al., 2021; Ferrario, 2019; Ferrario et al., 2024; Martino et al., 2019; Tanyaş et al., 2022; Tiwari et al., 2017). The cumulative effect of such earthquake sequences on rapid,

shallow landsliding is difficult to study as it requires satellite images to be acquired between each earthquake, but aftershock-triggered landslides can represent a considerable part of the total landslides for some events (Ferrario, 2019; Tanyaş et al., 2022). Unfortunately, in many areas of the world, multi-spectral satellite images are frequently obscured by clouds, preventing their use in landslide mapping for days or weeks at a time (Robinson et al., 2019). This can prevent differentiation between mainshock- and aftershock-triggered landslides and has been identified as a problem in many recent studies, for example on

the 2015 $M_w$ Gorkha, Nepal; the 2018 $M_w$ 6.9 Lombok, Indonesia; the 2018 $M_w$ 7.5 Papua New Guinea; the $M_w$ 6.8 Cotabato-Davao del Sur, Philippines earthquake sequences (Tiwari et al., 2017; Ferrario, 2019; Tanyaş et al., 2022; Ferrario et al., 2024). In addition, the reactivation or remobilisation late in the earthquake sequence of a landslide that failed early in the sequence may not be visible in medium resolution imagery such as Sentinel-2 or Landsat unless the shape of the scar noticeably changes.

Satellite synthetic aperture radar (SAR) data may offer a solution to this problem as these data can be acquired through

cloud cover and are sensitive to landslides. The Sentinel-1 SAR satellite constellation has acquired data every 6-12 days on two tracks globally since 2015. Recently several methods have been proposed to use these data to constrain the timings of shallow landslides (Burrows et al., 2022; Deijns et al., 2022; Fu et al., 2024; Wang et al., 2024). Here we apply SAR-based landslide timing methods to a sequence of six earthquakes that occurred over a one-month period in Lombok, Indonesia in 2018 in order to better characterise landsliding triggered during that event. We also demonstrate that for some landslides, an

InSAR coherence matrix approach can be used not only to constrain the timing of new landslides, but also to detect multi-stage failure such as reactivations (i.e. complete failure on one date followed by further failure within or connected to the landslide at a later date) and precursory motion (i.e. displacement on one date followed by complete failure of the same area at a later date). With this new information, we are able to draw conclusions on how landslide activity evolved during the 2018 Lombok earthquake sequence and discuss the implications this has for hazard and mass-wasting during earthquake sequences.

## 2 Data and Methods

### 2.1 Landslides triggered by the 2018 Lombok, Indonesia earthquake sequence

The 2018 Lombok, Indonesia earthquake sequence comprised 6 earthquakes of $M_w$ 5.8-6.9 between 28/07/2018 and 19/08/2018 (Fig. 1). These earthquakes occurred along the Flores Thrust Zone to the north of the island and triggered widespread shallow landslides across this area, particularly on the steep slopes of Mount Rinjani (Ferrario, 2019; Salman et al., 2020). The majority

of the study area is covered by tropical forest, with grassland areas at high elevations (>2000 m) and the uppermost part of the volcano covered by unvegetated volcanic deposits (Dossa et al., 2013). The majority of the population live along the coast.

Two landslide inventories have been published for the event, both identifying approximately 10,000 rapid and predominantly shallow landslides by the end of the sequence (Ferrario, 2019; Zhao et al., 2021). Ferrario (2019) also provide an inventory halfway through the sequence using imagery acquired on 08/08/2018, which allows the effects of the two largest earthquakes




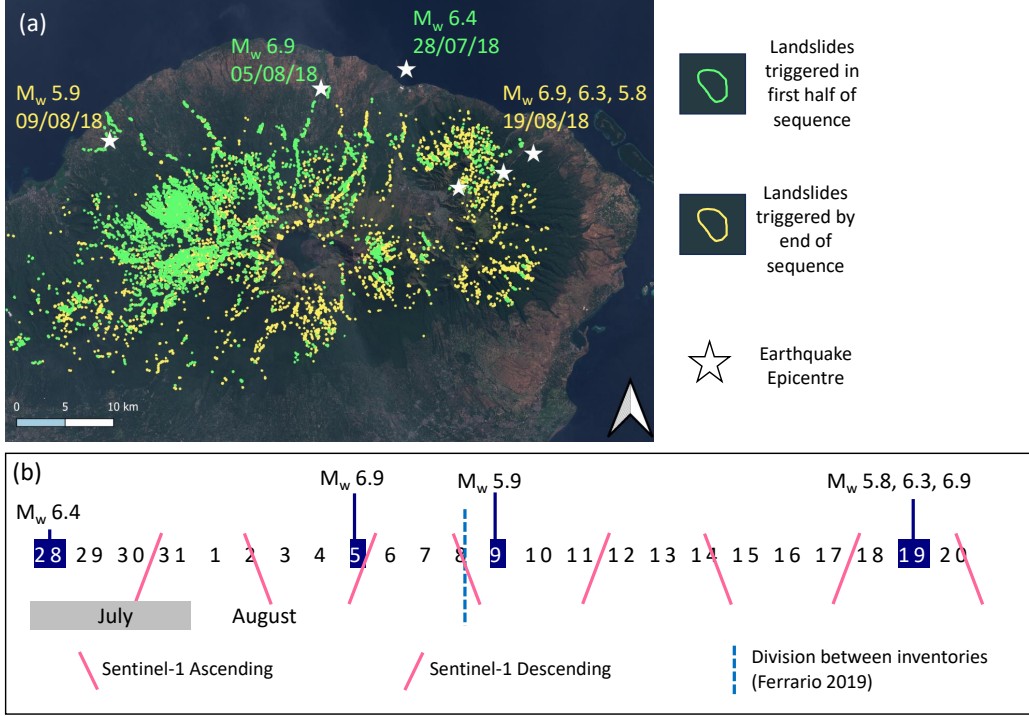

**Figure 1.** Landslides triggered in the first (green) and second (yellow) halves of the 2018 Lombok, Indonesia earthquake sequence as mapped by Ferrario (2019) overlain on Sentinel-2 imagery. Epicentral locations from USGS (2018). b) Earthquake dates and Sentinel-1 acquisitions throughout the earthquake sequence in July and August 2018.

(both $M_w$ 6.9), which occurred on the 05/08 and 19/08 to be separated. Zhao et al. (2021) provide a full inventory at the end of the sequence but were only able to generate a partial landslide inventory following the earthquake on the 05/08 due to cloud cover in the imagery used in that study. We therefore use the inventory of Ferrario (2019) in this study, which was generated manually through comparison of pre- and post-event PlanetScope satellite images.

       In both studies, cloud cover prevented landslide mapping following the first earthquake in the sequence ($M_w$ 6.4 on 28/07),
but Ferrario (2019), Zhao et al. (2021) and a preliminary report by Ganas et al. (2018) all agree that few landslides were triggered by this earthquake. The $M_w$ 5.9 earthquake on 09/08, which resulted in relatively weak shaking and would not ordinarily be expected to trigger many landslides (USGS, 2018c) was not included in these studies, but we include it here since similarly low magnitude earthquakes have been shown to increase landslide activity (e.g. Alfaro et al., 2012; Hallal et al., 2024; Martino et al., 2019, 2022). Furthermore, although this event was small in magnitude, it occurred at only 15 km depth and
preliminary analysis with SAR appeared to show some landslide activity at this time. Overall, it is expected that the majority of landslides failed during one of the two $M_w$ 6.9 earthquakes in the sequence on the 05/08 and 19/08, particularly the 05/08 event, which resulted in by far the strongest shaking (USGS, 2018b).





Ferrario (2019) mapped 4823 landslides (with a total area of 4.88 km$^2$) following the 05/08 earthquake and 9319 (10.25 km$^2$) at the end of the sequence (Fig. 1a). The high resolution (3 m) of the PlanetScope imagery used for the mapping means that

the inventory includes landslides with areas as small as 50 m$^2$. Such small events are unlikely to be resolved by the Sentinel-1 images used here, which have a resolution of 20 x 22 m and 60 x 66 m for backscatter and coherence images respectively (see Section 2.2 for details). For this reason, we limit the amplitude analysis in this study to 991 landslides > 2000 m$^2$ (following Burrows et al., 2022) and the coherence analysis to 371 landslides > 3600 m$^2$ (the size of the coherence window in Sect. 2.3).

## 2.2    SAR techniques for rapid landslide detection

Shallow earthquake-triggered landslides fail too quickly to be measured using differential InSAR techniques such as persistent scatterer interferometry, which require landslides to move less than a couple of cm between image acquisitions (Manconi, 2021). However, rapid landslides change the scattering properties and 3D shape of the Earth's surface, altering the amplitude and coherence of InSAR data. A large number of studies have explored the potential of these for detection of landslides in space (e.g. Burrows et al., 2019, 2020; Ge et al., 2019; Goorabi, 2020; Handwerger et al., 2022; Jung and Yun, 2020; Mondini,

2017; Mondini et al., 2021; Yun et al., 2015). Recently, several methods have been proposed to use coherence or amplitude to constrain landslide timings (Burrows et al., 2022; Deijns et al., 2022; Fu et al., 2024; Wang et al., 2024), taking advantage of the regular image acquisition strategy of Sentinel-1. Here we use both amplitude and coherence time series to constrain the timings of individual landslides triggered during the 2018 Lombok, Indonesia earthquake sequence and explore the potential of coherence to detect multi-stage failure.

## 85    2.2.1    SAR Amplitude

SAR images are acquired by active illumination of the Earth's surface by the satellite using microwave wavelength electro-magnetic energy. The amplitude of the signal returned to the satellite depends on the scattering properties of the material that this energy interacts with at the Earth's surface. The removal of vegetation and movement of material downslope alters these scattering properties as well as the 3D shape of the Earth's surface, giving landslides a signal in SAR amplitude maps (Mondini,

90 2017).

Two methods have been published that use amplitude data to constrain the timings of shallow landslides. Burrows et al. (2022) used step changes in time series of several amplitude metrics to indicate rainfall-triggered landslide timings. Fu et al. (2024) combined SAR and optical image time series to constrain the timings of 60 landslides, obtaining an average accuracy of around 23 days. Due to prevalent cloud cover in our study area and the fact that the landslides are already somewhat constrained

in time since the earthquake timings are known a-priori, we did not expect using optical imagery to offer an advantage here, so we used the method presented in Burrows et al. (2022). This method uses time series of four metrics: 1) the difference in average SAR amplitude for pixels within the landslide polygon compared to nearby similar "background" pixels; 2) variability between pixels within the landslide polygon; 3) geometric shadows cast by trees at the edge of the landslide polygon; and 4) geometric bright spots caused by dihedral scattering at the edge of the landslide polygon. Burrows et al. (2022) then identified



step changes in these metrics and used these to indicate landslide timings. These metrics, particularly those relating to geometric shadows and bright spots, work best in forested areas and can be applied to medium-large landslides ($> 2000 \text{ m}^2$).

The method was designed to be applied to rainfall-triggered landslides in monsoon climates where very little prior knowledge on landslide timing would be available. The case of earthquake-triggered landslides is somewhat simpler since we can assume that all landslides are concurrent with one of the earthquakes. Therefore, we slightly modify the method to make use of this information. For each of the four metrics $m$, we calculate the mean sum of the squares of the residuals $R$ if each earthquake is used to divide the time series into two sections $Y$ and $Z$ according to Eq. 1, where $\hat{m}$ represents the median value of $m$.

$$R = \frac{\sum_{i=1}^{n_Y}(m_i^Y - \hat{m}^Y)^2 + \sum_{i=1}^{n_Z}(m_i^Z - \hat{m}^Z)^2}{n_Y + n_Z} \tag{1}$$

The earthquake that minimises $R$ is selected as the most likely to have triggered the landslide. This is analogous to the approach used in clustering algorithms, where the data is divided in order to minimise within-cluster variance (Duda et al., 1973). This modification resulted in a small improvement in terms of accuracy and the number of landslides whose timing could be constrained compared to the original method (Fig. A1). As in Burrows et al. (2022), the more times an earthquake is selected (out of a maximum of 8: 4 methods x 2 tracks), the more confident we can be of the timing. Here, we require the same earthquake to be selected by a minimum of 3 metrics before it is accepted.

### 2.2.2 InSAR Coherence

InSAR coherence, which is derived from SAR amplitude and phase, can also be used to detect rapid landslides (Burrows et al., 2019, 2020; Goorabi, 2020; Yun et al., 2015). Coherence $\gamma$ is a measure of InSAR signal quality that is estimated for every pixel in an interferogram from its similarity to the pixels within a neighbouring window. This is described by Eq. 2 for an interferogram formed from two images $A$ and $B$ and a coherence window containing n pixels. $A_i$ and $B_i$ are complex representations of the phase and amplitude of each pixel i used in the estimation, with the overline representing the complex conjugate.

$$\gamma = \frac{\frac{1}{n}\sum_{i=1}^{n} A_i \cdot \overline{B_i}}{\sqrt{\frac{1}{n}(\sum_{i=1}^{n} A_i \cdot \overline{A_i} \sum_{i=1}^{n} B_i \cdot \overline{B_i})}} \tag{2}$$

In general, coherence is high when and where the acquisition conditions for the two images used to form the interferogram are similar. Changes in satellite position or Earth surface properties result in decorrelation. Landslides, along with soil moisture changes, movement of vegetation and other processes that alter the scattering properties of the Earth's surface result in low coherence. Previous works have observed InSAR coherence to (i) decrease for image pairs spanning the occurrence of shallow landslides (Burrows et al., 2019, 2020; Goorabi, 2020; Jung and Yun, 2020; Yun et al., 2015) and other forms of erosion and deposition (Bertone et al., 2019; Cabré et al., 2020, 2023; Liu et al., 1999) (ii) decrease for image pairs that capture precursory motion prior to catastrophic failures (Dini et al., 2022; Jacquemart and Tiampo, 2021) and (iii) increase for interferograms formed from post-event compared to pre-event image pairs due to the denudation of the hillslope by the landslide (Burrows et al., 2020; Deijns et al., 2022). This last effect reflects the strong influence of landcover type on coherence, with vegetated areas generally having a lower coherence than bare rock and soil in Sentinel-1 interferograms (Jacob et al., 2020).



Several studies have attempted to use coherence maps from consecutive pairs of SAR images to constrain landslide timings. The post-event coherence increase caused by hillslope denudation has previously been used to obtain the timings of seven very large (> 100,000 m$^2$) landslides (Wang et al., 2024) and to identify the timings of landslide inventories, in the case where it

can be reasonably assumed that all the landslides were simultaneous (Deijns et al., 2022). However, Wang et al. (2024) found that the method returned multiple possible failure timings for some landslides. Furthermore, when testing on individual, more moderately sized (> 2000 m$^2$) landslides, Burrows et al. (2022) found that pairwise coherence time series were too noisy to provide accurate landslide timings.

In order to improve the signal strength, here we increased the number of coherence maps used in the analysis by calculating

the coherence of every possible image pair in our time series. By taking the average coherence within a landslide polygon from every coherence map, we could then produce a full coherence matrix for each landslide (e.g. Fig. 2a). A previous study by Jung and Yun (2020) found this approach to perform poorly in forested areas, but their aim was emergency response, so they only used a single post-event SAR image. Furthermore, the method has been successfully applied to landcover mapping (Giffard-Roisin et al., 2022; Jacob et al., 2020), which suggests it should be able to detect at a minimum the denudation of the

hillslope caused by landslides. Finally, since previous studies have shown that coherence is sensitive not only to the denudation of the hillslope that can be captured by the amplitude method described in Sect. 2.2.1, but also to precursory movements and to movement of material in unvegetated areas (Bertone et al., 2019; Cabré et al., 2020; Dini et al., 2022; Jacquemart and Tiampo, 2021), the full matrix approach might be able to reveal multiple failure stages.

Figure 2a shows an example of a coherence matrix for a landslide in the Lombok study area that failed during the 05/08

earthquake. This is a square matrix of dimensions defined by the number of SAR images in the times series (e.g. 15 in Fig. 2). Diagonal elements are the coherence of each image with itself (i.e. maximum coherence = 1.0). Lower off-diagonal elements (x,y) record the coherence between the x$^{th}$ and y$^{th}$ image. Thus element (12,3) in Fig. 2a shows coherence between the 12$^{th}$ SAR image (acquired after all earthquakes) and 3$^{rd}$ (acquired prior to all earthquakes). Since the landslide occurred between these two SAR images, coherence is low. Element (13,12) shows coherence between the 12$^{th}$ and 13$^{th}$ SAR images. Since both

were acquired after the earthquake sequence had ended, and thus after the landslide had occurred, coherence is high. In Fig. 2a, coherence is generally highest whenever both images were acquired after 05/08 (matrix entries above row 9) and lowest for image pairs that span 05/08, with one image before and another after the earthquake (entries right of column 9 and below row 9). Pairs where both images were acquired before 05/08 (left of column 9) typically have intermediate coherence higher than those spanning the earthquake but lower than those after it. Upper off-diagonal elements are left blank because the coherence

map for image pair (x,y) and image pair (y,x) will be identical, so these elements would duplicate those already plotted.

Similarly to Sect. 2.2.1, we can automatically detect when the landslide failed by using each earthquake to divide the matrix into pre-event, co-event, and post-event sections, and identify which division minimises the residuals according to Eq. 1. We found several cases like Fig. 2b where the coherence matrix indicates that a landslide location has failed more than once. In this example, coherence is reduced for image pairs spanning 05/08, but is not consistently high after this event. Instead,

coherence is high for image pairs acquired after the 19/08 earthquake and is briefly high at element (11,10), where both images were acquired between the 09/08 and 19/08 earthquakes. This perhaps indicates three failures in this location: first on 05/08,




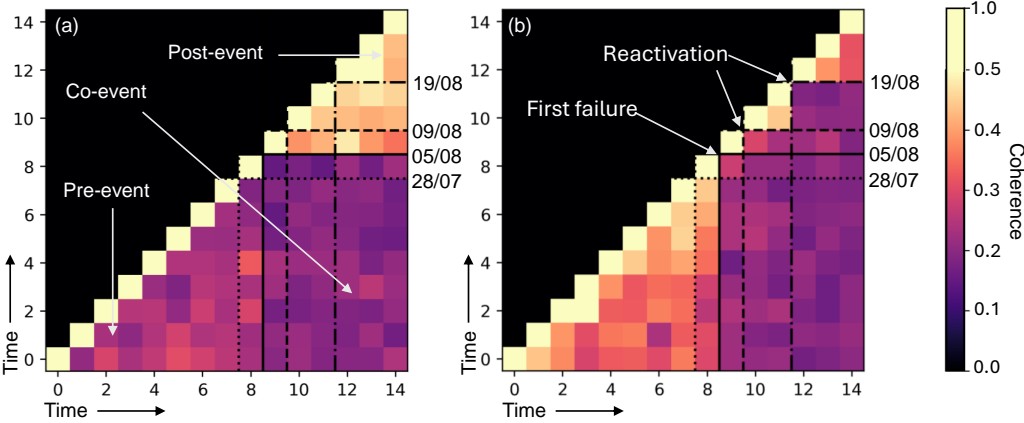

**Figure 2.** Coherence matrices for (a) a landslide in a vegetated area that failed during the 2nd earthquake (05/08/2018) (b) a landslide in an unvegetated area that initially failed during the 2nd earthquake (05/08) and then reactivated during the 3rd and 4th earthquakes (09/08 and 19/08). Black lines show the four earthquake dates in the sequence.

evidenced by reduced coherence for co-event relative to pre-event pairs; then on 09/08, evidenced by lower than expected post-05/08 coherence but high coherence post-09/08 and pre-19/08; and finally on 19/08, evidenced by low co-event and high-post-event coherence.

To make best use of this information, we carried out our analysis in two separate stages: first with the pre-event and co-event image pairs to identify the first failure and then with the co-event and post-event image pairs to identify the final failure. In order to estimate the quality of the selected division, we calculated the standard deviation of pixels in the co-event area and then calculated how many standard deviations apart the co-event and pre-event or post-event sections were. In the case where the ascending and descending tracks select different timings, we choose the one with the largest separation between the

sections. We chose a minimum threshold of 1.5 standard deviations in order to maximise the accuracy of the method. Raising this threshold beyond 1.5 reduced the number of timed landslides without improving the accuracy (Fig. A2).

## 2.3    SAR data and processing

Sentinel-1 collected images every six days on two tracks throughout the earthquake sequence (Fig. 1b). At least one ascending and one descending track image was acquired between each earthquake (with the exception of the three earthquakes all on

19/08). In both the amplitude and coherence analysis, we used images acquired over a 3-month period from 05/06/2018 – 05/09/2018. This amounted to 15 SAR images on the descending SAR orbit and 13 on the ascending orbit.

The amplitude analysis described in Sect. 2.2.1 uses ground range detected (GRD) images, which were accessed through Google Earth Engine following the method described by Burrows et al. (2022). These data have a resolution of 20 x 22 m and were used to calculate the four metrics listed in Sect. 2.2.1 for every landslide > 2000 m² in the inventory (991 events). For the



coherence analysis, Sentinel-1 single-look complex (SLC) images were processed using the GAMMA-based LiCSAR software
       package Li et al. (2016). SLC images were multi-looked by a factor of five in range, resulting in a resolution of 20 x 22 m
       (the same as the GRD product). Coherence was then estimated using a 3 x 3 pixel moving window, giving a resolution of 60
       x 66 m. No atmospheric correction was applied, since interferogram atmospheric effects are larger than the scale at which we
       processed our data Ding et al. (2008); Webb et al. (2020), and thus do not cause phase variations within the coherence window.
The coherence maps were then reprojected from the range-azimuth coordinate system in which SAR data are acquired to a
       geographic coordinate system. The average coherence within each landslide polygon > 3600 m$^2$ (371 events) was then obtained
       from every coherence map (estimated for every possible combination of SAR images) to generate the full coherence matrix for
       each landslide.

## 2.4   Validation of SAR methods against optical satellite images

In Sect. 3, we present the landslide timing results obtained from the SAR amplitude and coherence methods described in
       Sect. 2.2. In order to validate these results, we compare with the timing information that can be obtained from optical and
       multi-spectral images acquired during the earthquake sequence. This process is complicated by two factors.

       First, while our SAR methods have sufficient temporal resolution to assign a failure to a specific earthquake, this is not
       possible using optical images in all cases due to cloud cover. The inventories of Ferrario (2019) were generated using imagery
acquired after the earthquakes on 05/08 and 19/08, meaning that every landslide can be assigned to either the first or second
       half of the sequence. In some cases, we were then able to further constrain the timing using cloud-free areas of multi-spectral
       Sentinel-2 and Planet images and high-resolution optical images in Google Earth Explorer, but for around a third of the
       landslides, this was not possible.

       Second many landslides fail more than once during the sequence. To identify multi-stage landslides in the optical satellite
imagery, we initially compared the areas of polygons from the 05/08 and 19/08 inventories that overlapped and classed those
       which had increased in size between the two earthquakes as "multi-stage". Where this was possible, we then used the Sentinel-
       2 and Google Earth images to further constrain these changes in time. Landslides showing only small increases in area (< 100
       m$^2$) and cases where landslides were mapped as multiple polygons in one inventory but a single polygon in the other, were not
       classed as "multi-stage" as these differences could have arisen from differences in the images rather than landslide reactivation.
We also observed a small number of cases where the failed area after 28/07 or 05/08 in Sentinel-2 or Google Earth was better
       fitted by the 19/08 polygon than the 05/08 polygon in the inventories of Ferrario (2019). These cases were also not classed
       as multi-stage, since the discrepancy between the two inventories seems more likely to have arisen from mapping uncertainty
       (e.g. due to shadow in the Planet imagery) than from reactivation. Finally, we identified some cases of landslides initiating on
       28/07 and growing in size on 05/08, which were not initially classed as "multi-stage" because Ferrario (2019) did not map
landslides between these two earthquakes.

       Since both optical and SAR data can therefore yield multiple failure stages for a given landslide, a comparison between
       these two might agree, disagree or partially agree. Most statistical measures of performance, such as confusion matrices do
       not allow for partial agreement. Therefore, for landslides showing more than one failure in the optical data, we compare the



SAR timings against the optical result that seems the most relevant in each case. SAR timings derived from amplitude (Sect.
2.2.1), which primarily detect denudation of the hillslope were assessed against the timing of the largest failure by area in the
optical images. First and final failure timings derived from InSAR coherence (Sect. 2.2.2) were compared against the first and
last visible failures in the optical satellite imagery.

## 3 Results

### 3.1 Detection of landslide timings from SAR amplitude

After applying the SAR amplitude methods described in Sect. 2.2.1, we were able to constrain the timings of 307 of the 991
landslides larger than 2000 $m^2$. As the amplitude methods primarily detect denudation of the hillslope, for landslides that
undergo multi-stage failure, we expect the amplitude-derived timing to represent the main failure. Of these 307 landslides, 10
were assigned to the 28/07 earthquake, 269 to 05/08, 16 to the 09/08 and 12 to the 19/08. The timings that these 307 landslides
were expected to have based on optical imagery (Sect. 2.4) are shown in Table 1. Cells for which the two datasets agree are
in bold. Altogether, the optical and SAR timings agree for 269 (88%) of the landslides. Single failures had a higher rate of
agreement (91%) than multi-stage (81%). In Sect. 2.2.1, we required a minimum of 3 SAR amplitude metrics to select the
same timing before it was accepted. Burrows et al. (2022) found a similar accuracy (>90%) when imposing this requirement
and testing on three landslide inventories of known timing. However, the proportion of landslides timed in this study (30%) is
much higher than Burrows et al. (2022) were able to time at this level of accuracy (5-10%). Some of this improvement may
be due to the modification made to the method (Sect. 2.2.1, Supplementary information), but it is also likely to be due to the
tropical rainforest that covers most of our study area, since Burrows et al. (2022) found their method performed best in heavily
vegetated areas.

### 3.2 Detection of first and last failure timings from InSAR coherence

We were able to detect the first failure for 61 of the 371 landslides larger than 3600 $m^2$ using the coherence matrix approach
described in Sect. 2.2.2. Of these, 19 initiated during the earthquake on 28/07, 40 on 05/08, none on 09/08 and 2 on 19/08.
The SAR and optical timings agree for 49 out of 61 landslides (80%, Table 1). For the 12 cases for which the optical and SAR
disagree, the coherence matrix assigns an earlier first failure than the optical.

   We were able to detect the last earthquake a landslide failed in for 213 of the 371 events. This was the 28/07 earthquake in
4 cases (2%), 05/08 in 97 cases (46%), 09/08 in 30 cases (14%) and 19/08 in 82 cases (38%). This last failure timing refers to
245    the point after which there was no further failure by the end of our study on 5$^{th}$ September, so may correspond to a reactivation.
Overall, the two timings agree for 153 of the 213 landslides (72%, Table 1). Of the 60 cases where the two timings do not
agree, 32 are landslides that were mapped as failing only on 05/08 by the optical imagery, but have been assigned a later
final failure by the InSAR coherence. Thus, the coherence method has detected reactivations that are not visible in the optical
imagery. There is no way to further validate whether or not these were real events. Another 20 of these 60 are landslides that



**Table 1.** A comparison between timings derived from optical and SAR amplitude satellite data with each cell containing the number of landslides associated with a particular trigger. Cells where the two methods agree are given in bold font. Where possible optically mapped landslides were assigned to a causative earthquake (date columns) otherwise they are assigned to either 1st or 2nd half of the earthquake sequence (final two columns). The multi-stage failure timing derived from optical satellite imagery refers to the largest failure when compared to timings derived from SAR amplitude and to the first and last visible change when compared to first and last failure timings derived from InSAR coherence respectively

| | | Timing from Optical imagery | | | | | | | | | | | |
| | | Single Failure | | | | | | Multi-stage failure | | | | | |
| | | 28/07 | 05/07 | 09/08 | 19/08 | 1st half | 2nd half | 28/07 | 05/07 | 09/08 | 19/08 | 1st half | 2nd half |
|---|---|---|---|---|---|---|---|---|---|---|---|---|---|
| Timing from | 28/07 | **0** | 3 | 0 | 0 | **3** | 0 | **3** | 1 | 0 | 0 | **0** | 0 |
| SAR amplitude | 05/08 | 0 | **123** | 0 | 0 | **43** | 10 | 0 | **59** | 0 | 1 | **18** | 15 |
| | 09/08 | 0 | 2 | **3** | 0 | 3 | **3** | 0 | 2 | **0** | 1 | 0 | **5** |
| | 19/08 | 0 | 0 | 0 | **1** | 0 | **9** | 0 | 0 | 0 | **1** | 0 | **1** |
| First failure | 28/07 | **0** | 0 | 0 | 0 | **0** | 3 | **5** | 8 | 0 | 0 | **3** | 0 |
| timing from | 05/08 | 0 | **20** | 0 | 0 | **2** | 1 | 0 | **14** | 0 | 0 | **3** | 0 |
| InSAR | 09/08 | 0 | 0 | **0** | 0 | 0 | **0** | 0 | 0 | **0** | 0 | 0 | **0** |
| coherence | 19/08 | 0 | 0 | 0 | **1** | 0 | **1** | 0 | 0 | 0 | 0 | 0 | **0** |
| Last failure | 28/07 | **0** | 2 | 0 | 0 | **0** | 1 | **0** | 0 | 0 | 0 | **0** | 1 |
| timing from | 05/08 | 0 | **68** | 0 | 0 | **5** | 1 | 0 | **5** | 0 | 7 | **0** | 11 |
| InSAR | 09/08 | 0 | 10 | **0** | 0 | 0 | **3** | 0 | 0 | **0** | 5 | 0 | **12** |
| coherence | 19/08 | 0 | 22 | 0 | **7** | 0 | **6** | 0 | 0 | 0 | **15** | 0 | **32** |

were considered "multi-stage" based on optical imagery, having failed for the first time on 28/07 or 05/08 and grown in size by the end of the sequence, but which were assigned a final failure timing of 28/07 or 05/08 based on the InSAR coherence matrix. These are thus cases where the coherence has failed to detect reactivations that were expected based on the optical data. Possible explanations of the disagreement between the optical and InSAR coherence results are discussed further in Sect. 4.4. Overall, the coherence matrix approach appears to perform well, although relatively few landslides can be timed using it compared to the amplitude methods, in part due to the larger landslide size required for the coherence analysis.

### 3.3 Combination of amplitude and coherence to detect multi-stage failures and reactivations

Altogether for the 371 landslides > 3600 m$^2$, we derived the timing of first failure for 61 landslides and of the final failure for 214 landslides based on the coherence matrices. From the amplitude methods, we have timing information for 170 landslides, which we interpret as the "main" failure (whenever the most substantial denudation of the hillslope took place, since this is what the amplitude methods detect). How the timing datasets derived from coherence and amplitude overlap is shown in Fig. 3f. Overall, 258 of the 371 landslides are timed by at least one method, and 158 of these are timed by more than one method.





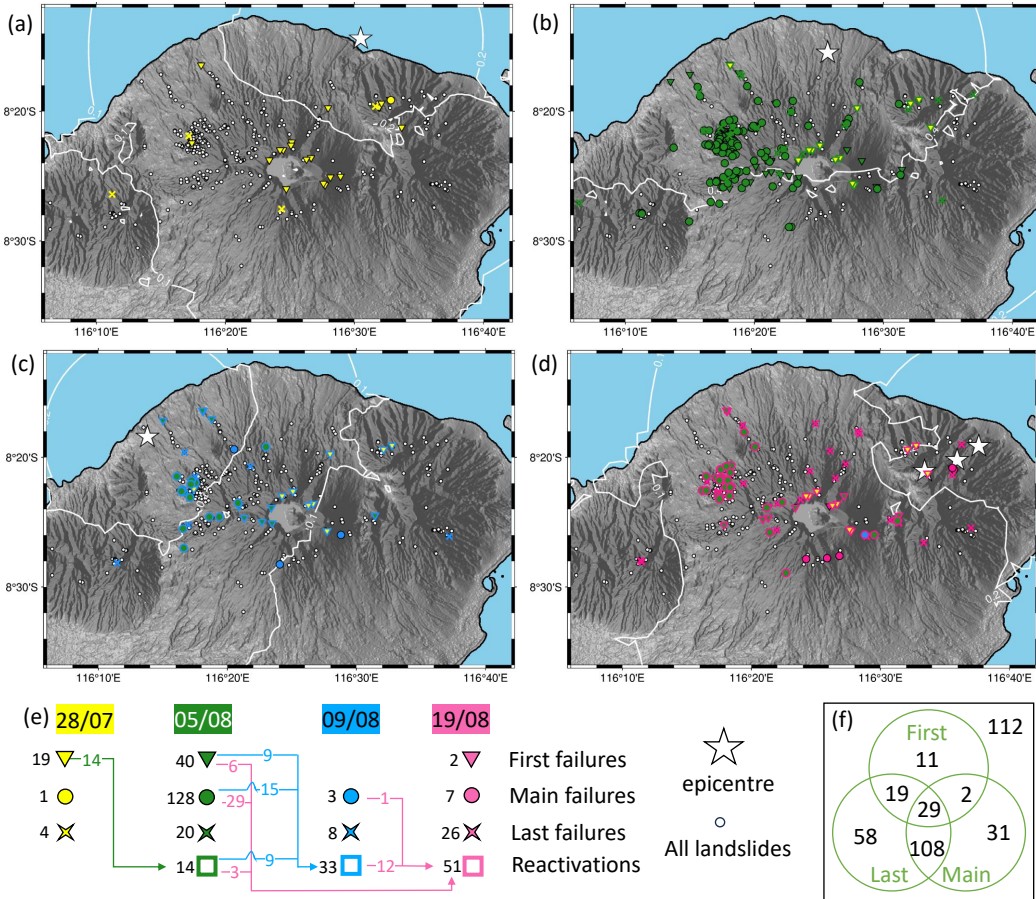

**Figure 3.** a) Landslides and reactivations timed with SAR following each earthquake in the sequence. Where first failure timing was available from coherence, this was plotted in preference over main failure. Last failures are only plotted when neither first nor main failure timings were available (since these would otherwise be reactivations). Reactivations are shown as highlights on the original failure. Modelled PGA contours of 0.1g, 0.2g and 0.4g from USGS are plotted as white lines. e) the number of each failure type after each earthquake with coloured arrows showing reactivations f) distribution of timing information available from SAR for the 371 landslides > 3600 m²

For 48 landslides, we obtained both the first and final failure timings from the InSAR coherence matrices. In 24 cases (50%), the two timings were the same, indicating a single period of failure. For 6 cases (13%), the first and final failures were associated with consecutive earthquakes. For 18 cases (38%), at least one additional earthquake occurred between the first and final detected failures. In these cases, we were able to visually inspect the matrix and identify that all 18 landslides were reactivated more than once, with 6 landslides (13%) active in all 4 earthquakes.

Although our amplitude methods provide only a single failure timing, they allow us to identify more examples of reactivations. When we have a main failure timing that is later than the coherence-based first failure, we can interpret the main failure as a reactivation (4 cases). When we have a main failure timing that is earlier than a coherence-based final failure, we




can interpret this final failure as a reactivation (45 cases). Reactivations are shown in Fig. 3 as coloured outlines around the original failure and by coloured arrows in Panel e. Landslide activity (derived from any method) was observed for a total of 259 landslides (Panel f) with activity at 24 sites (9%) on 28/07, 203 sites (78%) on 05/08, 44 sites (17%) on 09/08 and 86 sites (33%) on 19/08. These percentages total over 100% because some landslides are active in multiple earthquakes. In fact, reactivations made up the majority of the detected activity for the 09/08 and 19/08 events, 75% and 59% respectively (Panel
e).

## 4 Discussion

### 4.1 Triggering conditions for new landslides and reactivations

The landslide timing dataset we have generated allows us to make comparisons between the conditions required to trigger new landslides and those required to reactivate existing landslides. The occurrence of earthquake-triggered landslides is primarily
controlled by topography and ground shaking (Nowicki Jessee et al., 2018). Figure 4 shows the peak ground velocity (PGV) and slope at which new failures (Panel a) and reactivations (Panel b) occurred during the earthquake sequence. Estimates of PGV experienced during each earthquake were obtained from the USGS Shakemap webpage (USGS, 2018a, b, c, d). For 19/08, we took the maximum PGV experienced by each landslide during the $M_w$ 5.8, 6.3 and 6.9 earthquakes. In the majority of cases, this was the PGV of the $M_w$ 6.9 earthquake. Slope was calculated from the 30 m Copernicus digital elevation model
in Google Earth Engine and the maximum value was taken within each landslide polygon.

The landslide probability under these conditions can be estimated with the logistic regression model of Nowicki Jessee et al. (2018) using regression coefficients derived in that study for a global database of landslides. For lithology, we used the coefficient derived for intermediate volcanics, which comprise the majority of the study area according to the global lithological map of Hartmann and Moosdorf (2012) and for landcover, closed deciduous forest, which is the landcover type shared by most
of the landslides (Dossa et al., 2013). Although lithology and landcover also affect landslide susceptibility, we do not attempt to control for these: lithology does not vary much across the study area, particularly since many new landslides and reactivations occur on the same scars and so at the same locations. Differences in landcover between landslides is too difficult to account for since the landslides themselves mean that it changes through time.

Reactivations occurred at lower PGV and slope than new failures and a large proportion of reactivations occurred in locations
where the model of Nowicki Jessee suggests less than 1% probability of landslides (Fig. 4b). While the initiation of new landslides at the beginning of the sequence (28/07) occurred at PGV as low as 2.5 cm/s (PGA as low as 0.15g, Fig. 3a), this was mostly confined to slopes steeper than 35°(Fig. 4a). Later in the sequence, landslides on similarly steep slopes were reactivated at PGV as low as 1.5 cm/s during the earthquake on 09/08, while PGV values around 2.5 cm/s were sufficient to reactivate landslides on slopes shallower than 20∘ (Fig. 4b). New landslides initiating at such shallow slope angles were only
observed for the earthquake on 05/08, where the majority of landslides were triggered at PGV > 3 cm/s (PGA > 0.4g, Figs. 4a,3b).





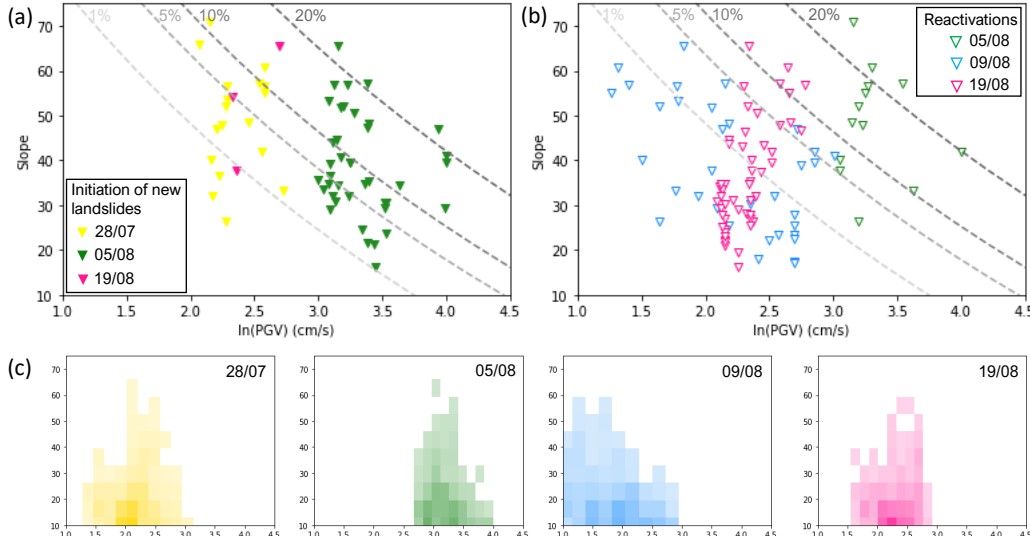

**Figure 4.** PGV and slope steepness at which (a) new landslides and (b) reactivations were triggered during the 2018 Lombok earthquake sequence. Dashed lines show the landslide probability under these conditions according to the empirical model of Nowicki Jessee et al. (2018) (c) 2D histograms of Slope and PGV across the study area during each earthquake.

This increased susceptibility to failure for reactivations compared to new failures is consistent with the increased levels of post-seismic rainfall-triggered landsliding that have been observed following many large earthquakes (Fan et al., 2021; Jones et al., 2021b; Marc et al., 2015; Tanyaş et al., 2021; Yunus et al., 2020). The mechanisms suggested for elevated susceptibility
to reactivation by rainfall are equally applicable to the case of seismic triggering studied here. They include damage to the regolith during the earthquake (loss of cohesion or internal friction); the loss of vegetation whose roots were contributing to the stability of the slope; steepening of the hillslope and increased presence of unconsolidated material in the form of co-seismic landslide deposits, which is easily remobilised (Marc et al., 2015; Fan et al., 2021). In other cases, such as the 2015 Gorkha, Nepal earthquake, increased post-seismic landslide susceptibility was driven by both the increased presence of (co-seismic)
landslide scars which were then able to be reactivated during the subsequence monsoon seasons (Dahlquist and West, 2019) and by more general, widespread damage to the landscape, which temporarily reduced the amount of rainfall required to trigger new landslides (Burrows et al., 2023). On the contrary, after the 05/08 earthquake, we observe reactivations at low PGV-slope combinations, but not new landslides (Fig. 4), indicating that here the primary driver of increased landslide activity is the increased presence of landslide scars rather than more general weakening of the landscape.

**4.2 Possible detection of precursory motion during the 28/07 earthquake**

18 of the 61 landslides for which we obtained the timing of first failure appear to have failed on 28/07. This proportion (30%) is very high compared to the results from amplitude, where the main failure was attributed to this earthquake in only 3 of



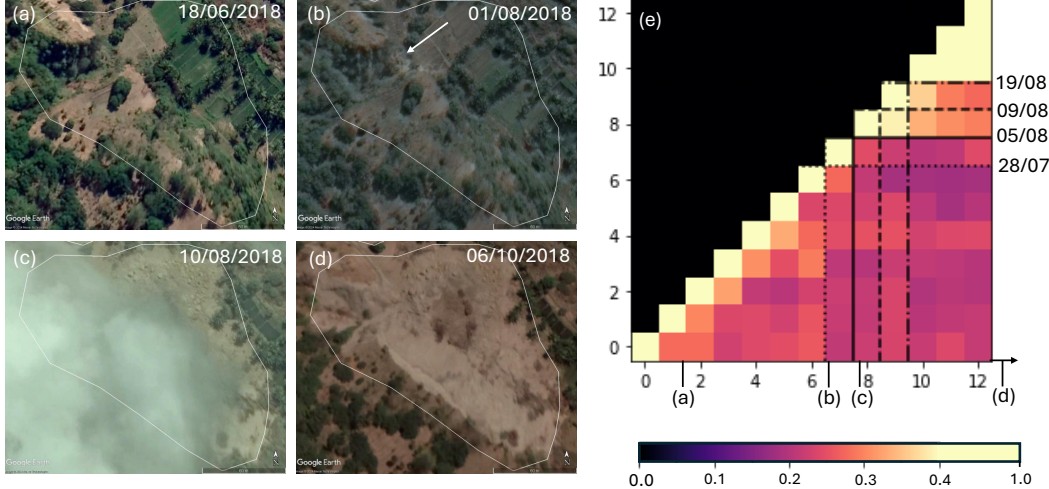

**Figure 5.** (a-d) Evolution through time of a landslide captured with high resolution google earth imagery acquired before (a), during (b,c) and after (d) the earthquake sequence. The white arrow on Panel b indicates a location where preliminary activity may have taken place following the earthquake on 28/07 prior to the main failure during the 05/08 earthquake (Panel c) (e) the coherence matrix for this event, showing landslide activity during earthquakes on 28/07, 05/08 and possibly 19/08. Panels a-d © Google Earth with white polygons from Ferrario (2019)

170 cases (2%). It also contradicts previous research based on optical satellite images, which found that in cloud-free areas, very few of the landslides triggered during the earthquake sequence occurred before the earthquake on 05/08 (Ferrario, 2019; Ganas et al., 2018; Zhao et al., 2021). One explanation for this discrepancy could be that some landslides exhibited precursory motion during the earthquake on 28/07 but did not fail completely until 05/08. Such motion would usually not be visible in optical satellite images or detectable using amplitude methods as it would not result in denudation of the hillslope. The short wavelength of Sentinel-1 (5.5 cm) means that coherence can be sensitive to relatively small movements and previous studies have demonstrated that small precursory movements can result in coherence loss prior to catastrophic failures (Dini et al., 2022; Jacquemart and Tiampo, 2021).

Figure 5 shows a possible example of this, where the landslide appears active during both the 28/07 and 05/08 earthquakes in the coherence matrix (Panel e), but large-scale failure is not visible in the optical satellite imagery until after 05/08 (Panel c). A white arrow in Panel b indicates an area where activity triggered by the 28/07 earthquake may be visible. However, these detectable changes are small, with trees and shrubs not perceptibly moved between 18/06 and 01/08 but clearly removed by 10/08. It is clear that no activity would be visible in lower resolution imagery until after 05/08 and so the landslide would not be mapped before this time. The SAR amplitude methods described in Sect. 2.2.1 assigned this landslide to the 05/08 earthquake, which is likely due to the fact that there is minimal change in landcover between panels (a) and (b) within the landslide polygon.





Interest in the potential to detect precursory movements prior to catastrophic failure with coherence or with displacements

derived from InSAR or optical image correlation, has grown in recent years as the acquisition frequency of satellite images has increased (Dong et al., 2018; Jacquemart and Tiampo, 2021; Lacroix et al., 2023). If coherence loss following the first earthquake in a sequence could be used to indicate areas susceptible to catastrophic failure during later earthquakes, this would be a useful risk management tool. However, we are only able to examine a very limited number of cases here. 14 of the 18 landslides that initiated on 28/07 were also active on 05/08 so could indicate precursory motion. Unfortunately, for the majority

of these, cloud free images were not available between the earthquakes on 28/07 and 05/08. Additionally, many took place in unvegetated areas where small movements would be challenging to see. This lack of vegetation also means that few (only 4) were also timed with amplitude methods. Thus overall, it is difficult to investigate this effect further here.

### 4.3 Implications for hazard and mass wasting

Several studies have attempted to draw associations between the shaking experienced during an earthquake or its magnitude,

and the likely severity of associated landslides Godt et al. (2008); Marc et al. (2017); Malamud et al. (2004); Nowicki Jessee et al. (2018); Tanyaş and Lombardo (2019). The results of our study highlight the complexity of this problem, since our dataset includes (i) landslides that did not fail until 19/08 despite experiencing stronger shaking on 05/08 (ii) landslides that did not fail on 28/07, but that were reactivated by comparable shaking on 09/08 or 19/08 after failing on 05/08 and (iii) landslides from 05/08 that were reactivated by 09/08 or 19/08 but not both, despite similar levels of shaking.

Overall, we believe that the earthquakes on 09/08 and 19/08 resulted in more landslide activity than they would have done had they not been part of the sequence. Although other sequences of relatively low magnitude earthquakes have triggered landslides, such as the 2011, $M_w$ 5.1 Lorca, Spain and $M_w$ 2020 Mila, Algeria events, both of which triggered over 250 landslides (Alfaro et al., 2012; Hallal et al., 2024), there are also cases where larger earthquakes have not resulted in extensive landslide activity, such as the $M_w$ 6.2 foreshock to the $M_w$ 7.0 Kumamoto earthquake (Xu et al., 2018). Our conclusion is further supported by the

fact that the earthquake at the beginning of the sequence (28/07) resulted in less activity than subsequent earthquakes on 09/08 and 19/08 despite having similar shaking intensity (Fig. 4c). 24 landslides were observed to be active in this event, compared to 44 and 86 respectively on 09/08 and 19/08. Thus, relatively weak shaking was required to trigger landslide activity later in the sequence, an effect which was also observed for aftershock-triggered landslides during the 2015 Gorkha, Nepal earthquake sequence (Tiwari et al., 2017). However, since this activity takes the form of reactivations rather than new failures, its spatial

extent is controlled by the shaking intensity experienced in the mainshock (Fig. 3c). This highlights the importance of rapid assessment of co-seismic landslides following a large earthquake as these can easily be reactivated by aftershocks.

It is difficult to determine the mass wasting effects of the multi-stage failure processes we have observed here with our coherence analysis. First, we identified more landslide activity than expected during the earthquake on the 28/07 (Figs. 3a, 4a). However, if these landslides would have failed anyway during the larger event on 05/08, the total mass wasting volume will be

unchanged. Equally if the earthquakes on 09/08 and 19/08 only resulted in downslope movement of unconsolidated co-seismic deposits left after 05/08, this material would likely have rapidly been remobilised by surface runoff or rainfall-triggered failure, so the overall effect on mass wasting would be minimal. If instead these earthquakes further damaged the rock and/or regolith





or caused landslide scarps to retreat, the mass wasting from the four earthquakes in sequence is likely to be greater than the sum of the mass wasting that would have been caused by each earthquake in isolation. For example, on its own, the earthquake on 370 09/08 would not be expected to trigger many landslides, (USGS, 2018c). Ferrario (2019) mapped many polygons that increased in size between 08/08 and the end of the sequence, suggesting that this may be the case, but we cannot differentiate between the two processes using SAR since both would result in coherence loss. Overall, while we are able to detect a mechanism that may result in increased mass wasting, different methods such as repeat LiDAR surveys would be required to fully quantify this process.

### 4.4  Disagreement between failure timings derived from optical and SAR datasets

When carrying out the validation of the SAR methods, cases were observed where the timings derived from SAR did not match the failures that were visible in the optical datasets (Table 1). While some of these may simply be due to inaccuracies in the SAR methods or manual landslide mapping, there are patterns that suggest that some of them may be explained by differences to what is and is not detectable in SAR and optical satellite imagery. For example, the 12 cases where the first failure timings 380 from InSAR coherence were early compared to the optical datasets could be explained if the InSAR coherence method has detected small, precursory motions that were not visible in Sentinel-2 or Planet imagery as in Sect. 4.2. This is particularly likely for the 6 landslides that were not visible until 05/08 in the optical imagery, but were detecting as failing on both 28/07 and 05/08 by the coherence matrix; and for the 4 landslides that were mapped in the second half of the sequence by Ferrario (2019), but were active in every earthquake according to the coherence matrix.

A similar explanation could be applied to the 32 cases where the landslide failed only on 05/08 according to the optical imagery, but whose last failure was detected on 09/08 or 19/08 by the coherence method. 29 of these 32 landslides were also identified as failing on 05/08 based on either InSAR coherence or amplitude (the other 3 were not assigned a timing using these methods). This suggests that these 29 last detected failures were reactivations rather than new failures. It is possible that these landslides reactivated without visibly changing the size or shape of the scar in the optical satellite imagery. Since InSAR 390 coherence is sensitive to erosion of unvegetated surfaces (e.g. Cabré et al., 2020), it might still detect such failures.

Finally, 24 landslide polygons were identified as increasing in size in the second half of the earthquake sequence based on the optical satellite images and thus classed as "multi-stage" (Sect. 2.4), but were assigned last failures with the InSAR coherence matrix that were too early. There are two possible explanations for this. First, it could be that the change in size or shape of the landslide polygon was due to differences in the Planet imagery and how shadows are cast at the edges of the forest rather 395 than reactivations. Second, if only part of a large landslide scar reactivates, it may change in shape in the optical imagery while most of the SAR pixels remain unchanged and do not lose coherence.

Overall, there are several differences between what can be detected with SAR amplitude, InSAR coherence and optical satellite images, which may be exacerbated by the different spatial resolutions of these data. The difficulty in detecting landslide reactivations in optical datasets, which is a motivation for developing techniques based on SAR is also a limitation when 400 validating these techniques.



## 4.5 Wider applicability and limitations of the coherence method

The coherence matrix approach we have used here has some specific advantages. First, it works well in unvegetated areas, where amplitude-based methods are less likely to return timing information (Burrows et al., 2022) and where landslide detection with optical data is particularly challenging. Second, using the coherence matrix approach, we are able to detect not only a single failure timing but also cases of precursory motion and reactivation. Reactivation of shallow landslides in earthquake sequences has been observed for other events, but studying this process requires either field photographs or high resolution multi-spectral satellite images, which are often obscured by cloud (e.g. Sepúlveda et al., 2010; Petley, 2024). The coherence approach used here could therefore provide a useful tool in studying this process. The ability to distinguish between active and inactive periods on shallow landslide scars could also be useful both in monitoring landslides that move too rapidly or are poorly oriented for differential InSAR techniques and in studying rainfall-induced reactivation of co-seismic landslides. The InSAR coherence approach we have used here could allow us to better study these processes using Sentinel-1 data or data from other SAR satellites with a regular acquisition strategy such as the planned NiSAR and ROSE-L missions (Jones et al., 2021a; Davidson and Furnell, 2021). However, there are other factors that can influence coherence that must be taken into account in future works, particularly soil moisture and InSAR spatial decorrelation (Scott et al., 2017; Kellndorfer et al., 2022).

### 4.5.1 Effects of perpendicular baseline on InSAR coherence

The overall coherence $\gamma_{total}$ can be broken down into three components according to Eq. 3 (Zebker and Villasenor, 1992).

$$\gamma_{total} = \gamma_{temporal} \cdot \gamma_{spatial} \cdot \gamma_{thermal} \tag{3}$$

Landslides, along with other processes that alter the scattering properties of the Earth's surface result in decorrelation of $\gamma_{temporal}$. Decorrelation of $\gamma_{spatial}$ resulting from small variations in the satellite orbit between image acquisitions can also result in coherence loss (decorrelation of $\gamma_{thermal}$ is caused by noise within the satellite receiving antenna and can usually be ignored). $\gamma_{spatial}$ is determined by the perpendicular baseline $B_{perp}$ (the distance between the satellite locations at the time the two SAR images were acquired), the difference between the SAR incidence angle $\theta$ and the local slope in the satellite line of sight $\alpha_{LOS}$, lightspeed $c$, the SAR wavelength $\lambda$ and chirp bandwidth $B_w$ and the sensor-target distance $r$ according to Eq. 4 (Lee and Liu, 1999).

$$\gamma_{spatial} = 1 - \frac{cB_{perp}}{\lambda r B_w}|cotan(\theta - \alpha_{LOS})| \tag{4}$$

Decorrelation of $\gamma_{spatial}$ is thus strongest for slopes that are close to the incidence angle of Sentinel-1 (32.9-43.1°) and face towards the sensor, an effect recently observed by Kellndorfer et al. (2022) who found that, for example, a slope at 30°would undergo decorrelation of 70% for an interferogram formed from two images with $B_{perp} = 81$m. Such values are not uncommon: in this study, the mean $B_{perp}$ was 53 m ± 40 m, while around 2/3 of the landslides occurred on slopes steeper than 30°(although not all of these were oriented towards the sensor). Decorrelation of interferograms formed with long $B_{perp}$ values was observed over many landslides in this study. Figure 6 shows one such example for a landslide in an unvegetated area



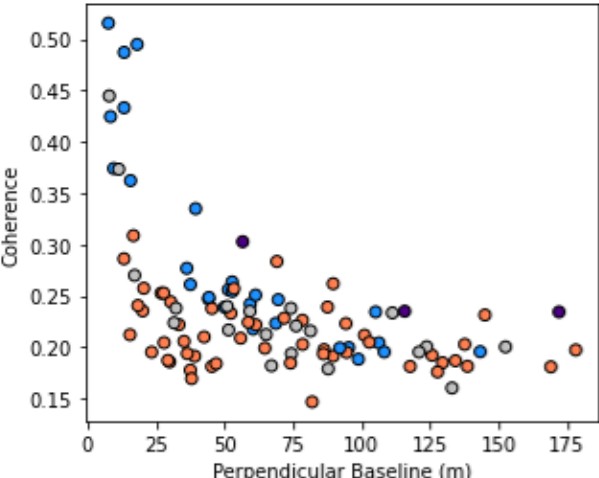

**Figure 6.** The effect of perpendicular baseline on coherence for a landslide in an unvegetated part of the study area

that slopes towards the satellite at an angle of 35.1°, so that, since $\theta = 35.0°$, $cotan(\theta - \alpha_{LOS})$ is close to 1. It can be seen that, while we are able to differentiate between co-event (unstable) and pre- and post-event (stable) periods for this landslide at shorter $B_{perp}$, this becomes difficult for baselines > 75 m. This underlines the value of the coherence matrix approach, since it allows us to more easily identify periods of low coherence caused by spatial decorrelation. Studies in arid environments have been able to normalise for this property (Liu et al., 1999), but it is more complicated here since the landslides result in a change in landcover type. In forested areas, decorrelation of $\gamma_{spatial}$ is compounded by volume decorrelation (Hoen and Zebker, 2000), but in Sentinel-1 interferograms, $\gamma_{total}$ is likely to be dominated by decorrelation of $\gamma_{temporal}$ in vegetated areas due to the movement of leaves in the canopy between image acquisitions (Jacob et al., 2020).

Overall, decorrelation of $\gamma_{spatial}$, which can be ignored for many applications, can have a strong impact in landslide studies, and should be considered in future works, particularly those that use different SAR constellations which may have longer $B_{perp}$ and those that go on to use Sentinel-1A in the coming years as $B_{perp}$ is likely to grow during this time (ESA, 2024). A Google Earth Engine tool to calculate $\alpha_{LOS}$ for an inventory of landslides and a given Sentinel-1 SAR scene, and so estimate the likely impact of spatial decorrelation for a particular event is available at https://doi.org/10.5281/zenodo.12579939

**4.5.2   Effects of soil moisture on InSAR coherence**

Changes in surface soil moisture alter the dielectric properties of the soil so can also decorrelate $\gamma_{temporal}$. A strong decorrelation signal has been observed for areas of bare rock and soil for Sentinel-1 interferograms formed from one wet and one dry image (Scott et al., 2017). Importantly, however, the decorrelation effect is not permanent. In the hyper-arid Atacama Desert, Cabré et al. (2020) were able to distinguish between changes in soil moisture, which are only temporary, and erosion, which

represents a permanent physical change. We can see a similar effect in Fig. 7, which shows a coherence matrix including a short rainfall event for an unvegetated area where a landslide occurred during the earthquake sequence.





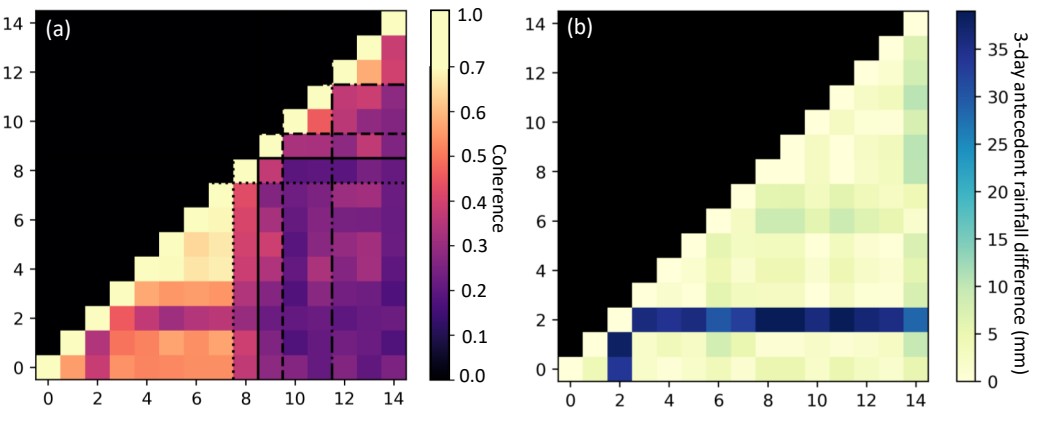

**Figure 7.** a) InSAR coherence matrix for a landslide in an unvegetated area previously affected by rainfall b) the difference in 3-day antecedent rainfall at the time of each image acquisition derived from GPM data (Huffman et al. 2015). The same rainfall event is visible in both matrices at the time image 2 was acquired.

While in this case, we are able to see that the soil moisture change was not permanent, the coherence loss was of a similar magnitude to that caused by landslide activity. This highlights the fact that there are some events for which coherence analysis may be inherently unsuitable, such as the case of an earthquake immediately followed by a storm, in which the coherence

signal will be the same for a co-seismic landslide scar that becomes wet during the storm and one that is reactivated. In this case, any drying effect would be hidden by the co-seismic failure. Snowfall, such as was seen following the 2023 Türkiye earthquake (Görüm et al., 2023), would also result in coherence loss and probably limit the applicability of the approach used here.

## 5    Conclusions

We have applied SAR amplitude and coherence techniques to characterise shallow landslide activity during the 2018 Lombok, Indonesia earthquake sequence. We have demonstrated that when a coherence matrix approach is used, we can detect not only single failures but also reactivations and thus build a more complete picture of landslide activity, although such methods cannot be applied to all landslides. Of the 177 landslides for which such analysis was possible here, 98 were active in more than one earthquake. In most cases these were reactivations, where failure in one earthquake was followed by further failure

in a later earthquake. However, in at least one case, our SAR techniques identified precursory activity prior to complete failure. This is consistent with theory and has been observed in a small number of previous studies but provides further encouraging evidence that at least some landslides may experience detectable displacement prior to full failure. Examining the drivers for slope instability, we found that new landslides generally followed pre-existing expectations of the shaking intensity and slopes associated with earthquake-triggered landsliding, but that reactivations of pre-existing scars required much less





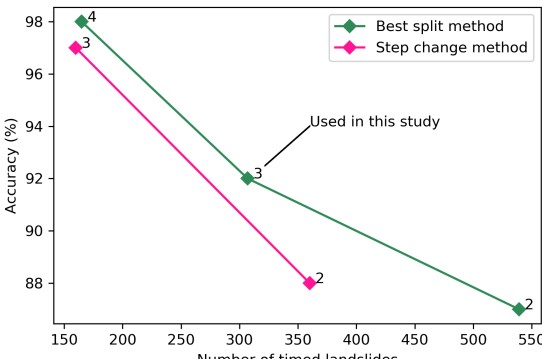

**Figure A1.** A comparison of accuracy verses for the original "Step change" method used by (Burrows et al., 2022) to identify landslide timings and the alternative "Best split" method used here (Sect. 2.2.1). The total number of landslides was 991 (all polygons > 2000 m$^2$ mapped by Ferrario (2019)). Accuracy was calculated for landslides whose timing could be constrained to specific earthquake using Planet, Sentinel-2 or Google Earth images. Points are labelled with the number of metrics required to select a time window for a landslide before it is accepted (When this number = 2, the metrics must be drawn from the same SAR orbit, following Burrows et al. (2022)

energy, occurring at accelerations as low as 0.1g. This demonstrates the difficulty in establishing predictive relationships for earthquake-triggered landslides. It also highlights the importance of rapid mapping of co-seismic landslide scars since these can easily be reactivated during aftershocks. Finally, the shift of reactivations but not new landslides to low PGV-slope combinations suggests that here it was the landslides themselves rather than more general landscape weakening that amplified landslide activity later in the earthquake sequence. This study represents one of the first combined applications of optical imagery and

Sentinel-1 amplitude and coherence to depict the multi-stage failure following a sequence of earthquakes. Application to other sequences of earthquakes or storms would require multi-temporal landslide inventories and good coverage with satellite images and would allow further confirmation and refining of the SAR methods.

*Code and data availability.* Sentinel-1 data are available from the Copernicus Data Space Ecosystem (https://dataspace.copernicus.eu). The original polygon landslide inventory is available in the supplementary materials of Ferrario (2019). Timing and reactivation information

derived from Sentinel-1 for this study will be published in an online repository following review. Computer codes for deriving landslide timings from SAR amplitude in Google Earth Engine are available at https://doi.org/10.5281/zenodo.6984291



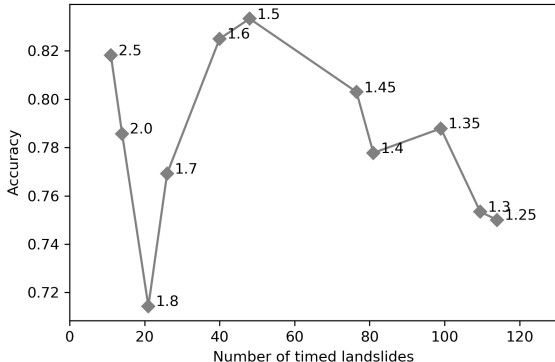

**Figure A2.** Accuracy of the InSAR coherence matrix technique when different thresholds are set for the number of standard deviations between "pre-event" and "co-event" for the timing to be accepted. Accuracy values are calculated only for landslides whose first failure could be constrained to a specific earthquake using Planet, Sentinel-2 or Google Earth imagery.

**Appendix A**

*Author contributions.* All authors were involved in the conceptualisation of the work, development of the methodology and preparation of the manuscript. KB carried out the data curation and analysis.

*Competing interests.* The authors declare no conflict of interest

*Acknowledgements.* SAR data presented here were processed during the PhD thesis of KB at Durham University under the supervision of Richard Walters, and we thank him as well as Thibault Taillade and Marcus Engdahl at ESA for useful discussions.

KB was funded by an internal research fellowship from the European Space Agency's Science Hub at ESRIN and by an MSCA postdoctoral fellowship (project id 101107490)



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
