# Peer review of "Detection of landslide timing, reactivation and precursory motion during the 2018, Lombok, Indonesia earthquake sequence with Sentinel-1"

_EGUsphere, 2024_

## Referee Comment (RC1)

Specific comments Abstract:

- ❖ To me it is not clear what methodology you have used. Is it a new methodology? And if so, did it work accurately? I think you could be more accurate in that.
- ❖ I think you can be more transparent in how representative the results are, especially considering that you only analyze a fraction of all landslides within the inventory. Are these results representative for the whole event?

Specific comments Introduction

- ❖ I believe that a more elaborated introduction on the SAR methodologies and the novelty of your work in regards to that would be relevant (around lines 36-39). Also, in the next line you mention InSAR coherence, but this has not been introduced yet (line 40).
- ❖ From line 39 onwards it starts to read like a conclusion, while I think a presentation of aim and objectives is more suitable. Is the aim to create a methodology to identify precursory and reactivation? Or, are you going to use old methodologies and identify their suitability in identifying that? Or, is the point more process-based and you want to understand the occurrence of landslides as a result of this earthquake sequence?

Specific comments Data and Methods

- ❖ To me it seems more relevant to discuss SAR data and processing (section 2.3) before the SAR detection methodologies (section 2.2). I think it is better to first properly introduce the SAR products and their properties, and how landsliding influence this signal before mentioning the detection methodologies. You have mentioned amplitude and coherence before, but what exactly consist of is unclear until section 2.3. I think a better structure and separation is relevant.
- ❖ Figure 1 could use some adjustments to increase readability:
    - o Inlay of location with respect to larger area needed.
    - o Fig. 1b could be improved visually. Needs x-axis + label. I find the forward and backslash a bit strange. Perhaps you can use a straight line with different colors, or patterns? In addition, I think a better distinction for the division in time between the two inventories for example by using different background colors (same as you use in fig1 a) would be relevant.
    - o Perhaps nice and informative to add an elevation map in Fig. 1a
- ❖ The reason for only taking landslides > 2.000m$^2$ during amplitude analysis and >3.600m$^2$ during coherence analysis is not well explained. Why 2.000m$^2$ where amplitude resolution is 20x22 m (440m$^2$) and 3.600m$^2$ where coherence is 60x66m (3.960m$^2$). This means that coherence resolution is lower than some landslide events?
- ❖ Following previous point: Particularly for coherence you have a rather low resolution. How does this influence the detection results? Does this mean that some landslides are only covered by one pixel? How does that work with mixed pixels? This could be the case for many landslides right? How does this influence your methodology? This is not really mentioned. In addition to that, I believe that presenting the size distribution of your inventory is important for the interpretation of the results and their accuracy.
- ❖ I'm also curious why you use a low coherence resolution. A moving window not necessarily reduces the resolution of coherence?
- ❖ Since you are proposing a new application of a methodology using the coherence I think this requires some sort of a sensitivity analysis. Given that the land cover is more or less the same, it would be interesting to see the accuracy in regards to size.

- ❖ In the end you use less than 10% (after size threshold) of the complete inventory. I think this will affect the interpretability of the results, but this is not really mentioned specifically.
- ❖ How it is currently written, I do not fully agree with your reasoning to not use optical data (line 96) in the methodology. Cloud cover does not necessarily have to cover the full event completely. There could always be some cloud free spots, especially in regards to earthquake-triggered landslides (where rainfall doesn't play a major factor). This information can be used and does not necessarily rule out the use of optical.
- ❖ Rainfall could also play a role in influencing the amplitude and coherence values. Was there any (heavy) rain during the sequence that could have influenced the results?
- ❖ For figure 2. It seems like the coherence of element (13,14) that indicates post-event stable conditions and element (9,10) that indicates reactivation are almost similar to each other? What does that mean?
- ❖ What polarization are you using and why? I think this needs some elaboration.

Specific comments Results

- ❖ To me it is rather unclear if (and for which landslides) the SAR-based precursory or reactivation conclusions are validated using optical imagery. Now it reads like this is not really the case. Can this difference in timing be explained by inaccuracies of the methodology? Potential noise? Basically, how reliable are these results? I would like to see a figure where it becomes clear the reactivation or precursory movement defined by SAR products is in fact actual precursory movement or a reactivation. From table 1 to me this doesn't become really clear.
- ❖ You mention that you are only able to derive only a portion of the landslides. Why is this? I think this is relevant needed information.
- ❖ I have to say that I was a bit confused by the mentioning of all these landslide timing detection numbers throughout sections 3.1-3.3. A visual would probably help with the interpretation. Maybe a better back and forth with figure 3 could benefit this?
- ❖ As a suggestive question: In the end I wonder about the use of SAR if there is only ~30% (of the 10% of the total inventory) that can be detected. Were you able to identify more accurately using optical (even if it is manual of course)? Perhaps there is many landslides that now have a more accurate timing using optical than SAR? Mentioning this could increase transparency in regards to the applicability of the methods used.
- ❖ You seem to assume that when the timing is done, the estimation is 100% correct. The reasoning for this is not clearly explained, what is the uncertainty in this? For example, in line 248 you mention with certainty that they have to be reactivations. Is this really true? No noise at all? Can it just be that the timing detection is wrong? Especially considering that the coherence doesn't always work correctly as you mention 252-253. I think this is an important aspect to address.
- ❖ Fig 3:
  - o Fig. 3f to improve readability maybe increase the size of the circles proportionally
  - o Fig. 3f, should 112 be 113? Then 258 adds up to 371 as mentioned in line 261
  - o I think you should add a reference to the PGA data of the USGS

Specific comments Discussion

- ❖ The first two paragraphs under line 278-293 would probably better fit in the method section rather than discussion. In addition, the rest of section 4.1 would better fit in the results section.

- ❖ Fig 4:
  - ○ Fig4c legend needed, what do the colors represent?
  - ○ Fig4c why not subdividing them into new and reactivations as well?
  - ○ the yellow color is not properly readable
- ❖ I think the representativeness of the results should be discussed. You results do not include smaller landslides and they only consist of a fraction of all the events.
- ❖ Figure 5:
  - ○ Adding a scale bar seems essential for the interpretation in relation with coherence resolution.
  - ○ No x-label and y-label for 5e, and no x-label for the legend
- ❖ In line with previous comments, I'm not entirely convinced that this coherence change indicates precursor movement, my main concerns being:
  - ○ How does the resolution affect these processes, especially given that you have quite a low coherence resolution? Also, how does the mixing signal in pixels influence this?
  - ○ Isn't there any uncertainty in the coherence based timing?
  - ○ Wouldn't you need a longer time series, to derive a trend (like Dini et al., 2022 and Jacquemart & Tiampo, 2021)? Perhaps it is just a noisy image? Maybe there is still some effect of local clouds, or local rainfall?
  - ○ Figure 5:
    - ▪ This figure would greatly benefit from adding a coherence map. Then we can see the coherence response, and even see how the coherence pixels cover the landslide location. Now it is not transparent.
    - ▪ From images 18-06-2018 to 01-08-2018 you can see that some of the grasses have been removed. Can the coherence loss be attributed to this vegetation change?
    - ▪ If the precursory movement is due to the 28/07, why is the coherence rather low the image before this event as well (element (5,6))? Or, am I interpreting it incorrectly?
- ❖ NISAR will have a different wavelength than Sentinel-1. This can impose differences and alter the usability of your methodology. This might require elaboration.
- ❖ How has this perpendicular baseline effect affected your results? Could it have induced noise?
- ❖ Were there any rainfall events in your study area during the earthquakes that have affected the results?
- ❖ Figure 6:
  - ○ Meaning of the color? Legend required. Which image pairs do these dots indicate?
- ❖ Figure 7:
  - ○ When did this event occur? Why is there little more consistent higher coherence post-event?
  - ○ There is little explanation on Fig. 7b. I think it could use some more elaboration.
  - ○ X- and y-labels needed for both subplots

Main comments Conclusion

- ❖ You mention that this study combines optical and SAR. However, this is not what you mention in the introduction and methods. You use optical as validation, right?

Line-by-line comments

Line 3: Adding a timeframe in which the sequence occurred here would be relevant

Line 5/6: Here you use 'many' a lot, I think you should be more accurate and present some percentages.

Line 8: I find this sentence slightly unclear 'weakening effect' of what? Could probably benefit from some rephrasing

Line 13-14: I find the meaning of 'significant mass wasting effect' to be a bit unclear

Line 15: Possible addition: 'In particular, earthquake-induced landslide inventories' to be more precise in which type of landslide inventory you are addressing.

Line 24: What do you mean by: 'cumulative effect'?

Line 24-27: I find this sentence to be a bit unclear. What is the point of this sentence? I would advise to rephrase for clarity.

Line 31: The references here are not ordered the same as the others, maybe better to add the relevant reference after each earthquake?

Line 32-33: The point you mention about medium resolution not being able to capture reactivation or remobilization is not that clear. It now reads like SAR will be able to provide a solution, but how will SAR be able to fix this while the resolution is lower?

Line 60-61: 'few landslides were triggered by this earthquake'. I'm wondering where this statement is based on if there is no imagery available. I think it will be helpful to clarify that.

Line 68: Does these number total to ~15.000 or are the inventories overlapping? It is slightly unclear.

Line 72-73: It is not fully clear if these 991 and 371 landslides are only due to this size threshold, or if there were other factors on which the reduction is based?

Equation 2: Slightly unclear what it does, what is i, what is n?

Fig. 2: I'm wondering if it might be more useful to add the actual dates to get a better understanding of the temporal baseline as well?

Line 75: Differential InSAR comes a bit out of the blue and maybe should be introduced first?

Line 82-84: Wouldn't this sentence be more appropriate at the end of the introduction? Here you identify that it still has an exploratory aspect.

Line 88-90: I think soil moisture should be added here as well, since it influences the amplitude values.

Line 91: This seems to contradicts your point in line 81 where you mention that there are multiple methodologies for this.

Line 103: 'very little prior knowledge on timing': Although this is a relative statement, I would argue that you have rather a lot of information already, a few months accuracy. I would instead mention this few months accuracy instead of 'very little'.

Line 104: 'are concurrent with one of the earthquakes': I'm not an expert in earthquake related landslides, but are there no landslides that occur a few days after the shock? There is no doubt whatsoever?

Line 148: I think it is a bit unclear how this full matrix approach might be able to do that. I think some elaboration is required.

Line 155-160: Why is there a difference between the coherence values before and after the earthquake? You use typically and generally, but it is unclear why. For better interpretability I would relate these values to the actual landscape conditions.

Line 163: 'has failed more than once': I'm curious how certain this is? Can it just be noise?

Line 170-176: I have to admit, that this part is a bit unclear to me. I think this section would benefit some from elaborating a bit more on 'our analysis' and how this relates to equation 1.

Line 195: This first sentences seems unnecessary

Line 201-203: This use of optical data contradicts your initial statement that optical data is not relevant for timing detection here. I think that requires some rephrasing

Line 204: 'Second, many landslides fail more than once': How do you know this? This has not really been clearly presented in the inventory section.

Line 211: 'fitted' what does this mean? Little unclear

Line 227: 'represent the main failure' Why is that? I think this could use some elaboration

Table 1: you use 05/07 instead of 05/08 for optical timing

Line 239-244: While you present the same type of results, you do it differently (with/without percentages/ different way of telling it). I think it is better readable if there is a similar structure.

Line 246: At first, this number (153 of the 213) confused me a little bit since you first say 61 and then 213, but I see now that you mean difference between optical and SAR. I think this should be clarified.

Line: 258: '214' should be 213, right?

Line 258: '170' where does this value come from? In sect 3.1 you mention 307. Or, is this only for the >3600 m2 landslides?

Line 260: First mention of the figure 3 is Fig 3.f shouldn't it be Fig. 3.a then?

Line 265-266: I think this statement requires backup in the form of results or supplementary material for transparency.

Line 272: Is it 259 or 258 (as in line 261).

Line 296: '2.5cm/s' Even lower right? 2.1/2/2

Line 299: degree sign needs to be changed

Line 345-346: References should be between brackets? Ending with a full stop

Line 350-351: I think this statement should be backed-up by some reasoning.

Line 359-360: This sentence seems a bit unclear? Don't you mean to say that the spatial extent is already defined by the landslides that occurred in the main shock? Maybe some rephrasing is needed.

Line 362 & 367: What do you mean by ' mass wasting effect'? amount of regolith mobilized?

Line 381-384: You say it is 'particularly likely' but why is that?

Line 388: Instead of reactivations, could noise be a factor?

Line 444: I think this should be included as a reference, right?

---

## Author Comment (AC1)

**Response to reviewer 2**

I find the manuscript interesting, and the topic and experiments are relevant. Overall, I believe the manuscript would benefit from some clarifications and restructuring to enhance clarity and flow. Below are some suggestions for improvement:

- The manuscript presents two key novelties: (i) the coherence matrices method and (ii) the novel insights into the specific earthquake sequences that this method, along with the amplitude-based approach, helps uncover. However, while the introduction focuses more on the method (i), the conclusions emphasize the second (ii) novelty. Aligning these sections more closely could strengthen the manuscript.

  Following this comment and comments from reviewer 1, we have made the following change to the introduction to better outline the aims of the paper

  *We also demonstrate that for some landslides, an InSAR coherence matrix approach can be used not only to constrain the timing of new landslides, but also to detect multi-stage failure such as reactivations (i.e. complete failure on one date followed by further failure within or connected to the landslide at a later date) and precursory motion (i.e. displacement on one date followed by complete failure of the same area at a later date).*

  Changed to:

  *We use the amplitude-based method of Burrows et al. (2022) to constrain the failure timing of new landslides. We also explore an approach based on interferometric SAR (InSAR) coherence matrices, a technique that has successfully been applied in landcover mapping (Giffard-Roisin et al. 2022; Jacob et al. 2020), but not yet tested for landslide timing. Here, we identify landslides where this method appears to identify multi-stage failure such as reactivations (i.e. complete failure on one date followed by further failure within or connected to the landslide at a later date) and precursory motion (i.e. displacement on one date followed by complete failure of the same area at a later date).*

  In the conclusions, we have then made the following change to better highlight the novelty of the method as you suggest.

  *"We have demonstrated that when a coherence matrix approach is used, we can detect not only single failures but also reactivations and thus build a more complete picture of landslide activity, although such methods cannot be applied to all landslides."*

  Changed to *"We have assessed a new method for landslide timing detection based on InSAR coherence matrices. This approach, which is mainly applicable to larger events, can detect not only single failures but also reactivations and thus build a more complete picture of landslide activity."*

- The method could also be valuable in distinguishing landslides triggered by different rainfall peaks occurring close in time (e.g., Emilia Romagna in 2024). As you note, this introduces additional complexities, but it could be an interesting avenue for future research. The University of Bologna has open-sourced a highly accurate dataset for that event, which may be useful for further exploration.

  While this is true, we believe that more testing would be needed for the InSAR coherence matrix before it can be applied to rainfall events due to the sensitivity to changes in soil moisture detailed in Section 4.5.2.

New text at line 458 of the original manuscript *"Because of this sensitivity to soil moisture changes, further testing is needed before the InSAR coherence methods can be applied to rainfall triggered landslides. Landslides triggered by sequences of storms are also often poorly constrained in time, and while the amplitude-based methods can be applied in this case (as in Burrows et al. 2023), InSAR coherence could also be beneficial in study landslide reactivation and landslides in unvegetated areas."*

- The choice of thresholds for discarding landslides (<2000 m² for the amplitude approach with a 22x20 pixel size and <3600 m² for the coherence approach with a 60x66 pixel size) should be better justified. Additionally, it would be helpful to clarify why the amplitude-based approach includes landslides approximately five pixels in size, while the coherence-based approach includes landslides as small as a single pixel.

  For the amplitude methods, it is necessary to have multiple pixels within the landslide polygon to calculate the metrics (e.g. pixel variability) used by Burrows et al. (2022). Therefore, the polygon needs to be larger than 22 x 20 m2. An analysis of the effect of landslide size on the sensitivity of the method was carried out in that paper.

  Text changed at line 101 of the original manuscript from *"These metrics, particularly those relating to geometric shadows and bright spots, work best in forested areas and can be applied to medium-large landslides (> 2000 m2)."* to *"These metrics, particularly those relating to geometric shadows and bright spots work best in forested areas. The method is limited to landslides > 2000 m2 so that each polygon contains enough pixels to calculate metrics (e.g. pixel variability) and is more sensitive for larger landslides Burrows et al. (2022)."*

- If landslides below 2000 m² and 3600 m² are discarded because they are assumed to be too small for detectable changes, how does this impact the detection of reactivations? Are reactivations generally larger than these thresholds, or do you believe smaller reactivations can still be detected?

  This is variable since a SAR pixel can be dominated by a single object and if that object moves, this could result in coherence loss. With the optical data we are using to verify the SAR methods, it is not possible to see how much of a landslide scar has reactivated, only whether or not the denuded area has increased in size. See our response to your next comment for the change we have made to the text.

- Regarding reactivations, do you primarily detect an increase in landslide area, or do you also observe failures within the existing scar?

  Failures within an existing scar are not visible in the optical imagery, making it difficult to verify when we do and do not detect them. While we agree that this is an important thing to test, it would require data that we do not have for this event. In response to this and your previous comment, we have added the following sentence to Section 4.4 (Line 390 of the original manuscript):

  *"To confirm that these are indeed reactivations, comparison against a different dataset, such as field surveys, ground-based SAR or high resolution DEMs would be necessary, but this is beyond the scope of this study."*

- How do you account for geometric distortions? For example, do you remove shadowed areas a priori, or do you rely on the assumption that using both orbits provides a high probability of capturing meaningful data?

  The code described in Burrows et al. (2022) automatically applies a shadow and layover mask to the data as part of the amplitude methods. We have not applied one to the coherence methods, but landslides that are affected by shadow and layover in one SAR orbit will be noisy and susceptible to geometric decorrelation. Therefore if they are assigned a timing based on coherence, it will be from the other orbit.

- Figure 2 is crucial but could be made clearer. A suggested improvement would be to plot coherence on the y-axis and real dates on the x-axis, with coherence values represented as horizontal lines extending from the date of the first image to the last. Not sure this would help, but it is worth a try.

  We tested the style of plot you suggest, but feel that the division into co-event, pre-event and post-event by minimising the residual is easier to understand if the coherence data are displayed as they are. In addition, plotting the coherence as a matrix is how it has been done in various other studies which use coherence for other applications (Jung and Yun, 2020; Giffard Roisin et al. 2022; Jacob et al. 2020 – full citations in manuscript) and we want to be consistent with this body of literature. However, we have improved Figure 2 by splitting it into two figures where panel (a) is now shown alongside coherence maps and panel (b) is shown alongside time series of multispectral imagery that show multi-stage failure

- The validation of this approach is thorough and well-executed. However, I have some questions regarding the terminology. In Section 2.4, you state that optical and SAR data can agree, disagree, or partially agree. Could you expand on what "partially agree" means? Additionally, framing it this way implies there is no ground truth. However, in cases where high-resolution optical imagery confirms a reactivation linked to a specific shaking event, wouldn't that be considered ground truth? Even a few well-validated cases could be sufficient to support the analysis and conclusions.

  By partially agree, we meant, for example, the case where we think the landslide failed on 05/08 and 19/08 based on the coherence, but from the optical, the landslide failed in 05/08 only. This would mean the two datasets partially agree (that the landslide failed during 05/08) but partially disagree (over whether or not the landslide failed on 19/08).

  To clarify this, we have changed the text at line, by adding the text in bold font at line 217 of the original manuscript:

  *"Since both optical and SAR data can therefore yield multiple failure stages for a given landslide, a comparison between these two might agree, disagree or partially agree (i.e. in the case of multi-stage failure, agree for one detected failure timing, but not for another)"*

  To better support the analysis, we have divided the old figure 2 and, for the coherence matrix example showing the reactivation, we have added multi-spectral planet imagery showing the landslide failing during EQ2 and then growing in size during EQ4. There are thus two examples shown in the paper: this new figure and figure 5 of the original manuscript.

- It would be helpful to explicitly state that the conclusions are valid for landslides above a certain size threshold, as smaller failures may behave differently.

In response to this and comments from reviewer 1, we have added an analysis of how landslide size affects the sensitivity of the InSAR coherence to section 3.2 as follows

*"Within the landslides examined, larger landslides were more likely to be assigned a timing by the InSAR coherence methods than smaller landslides. The inventory of Ferrario contained 87 landslides > 10000 m2, 38 in the range 8000-10000 m2, 75 in the range 6000-8000 and 171 in the range 3960-6000 m2 of which 70 (80%), 25 (66%), 46 (61%) and 86 (50%) were assigned a timing respectively."*

- In the abstract, I recommend opening with the importance of attributing landslides to specific triggers to highlight the study's relevance.

  In response to this comment, new text has been added at line 2. *"This information is crucial for understanding their triggering conditions."*

- Line 11: "sequences of triggers" → "sequences of earthquakes." "sequences of triggers" has been changed to "earthquake sequences" in the revised manuscript**.**

- Section 2.2 could be removed, as it does not seem directly relevant to the study. Lines 75-84 have been removed from the revised manuscript

- Line 103: "somewhat simpler" → Consider rewording to "We modify the approach as we can assume landslides occur concurrently..." for clarity.

  Thank you for your suggestion

  *"The case of earthquake-triggered landslides is somewhat simpler since we can assume that all landslides are concurrent with one of the earthquakes. Therefore, we slightly modify the method to make use of this information."*

  Has been changed to *"Here we modify the method since we can assume that all landslides are concurrent with one of the earthquakes."* In the revised manuscript

- Line 465: Could you clarify what kind of precursory activity you are referring to?

  Yes we have rewritten this sentence to clarify our meaning

  *"However, in at least one case, our SAR techniques identified precursory activity prior to complete failure."* has been changed to

  *"However, in at least one case, our SAR techniques identified precursory activity: small movements during one earthquake in an area that then failed during a later earthquake."*

---

## Author Comment (AC2)

**Responses to reviewer 1 comments**

This manuscript provides an interesting research into the use of Sentinel-1 SAR data to accurately identify the timing of earthquake-triggered landslides, their reactivations and potential precursory motions. The authors apply a combination of amplitude and coherence-based timing detection methodologies on the 2018 Lombok, Indonesia earthquake sequence which results in a multi-temporal inventory. This then allows them to interpret the different triggering conditions of new and reactivated landslides.

I believe this study is very interesting, the techniques are very relevant, and could have potential to be adapted to also identify timing of rainfall-triggered landslides. However, I think this manuscript could benefit from additional considerations on the methods and writing. My specific comments and line-by-line comments are attached as supplement. These comments mainly come down to the following main points:

- The objectives of this paper are not that clear from the introduction. I believe that the introduction would benefit from a clear presentation of the aims and objectives of the manuscript. Currently it does not become clear that the coherence matrix approach is a novel technique in this context which you are going to explore the usage of. This has implications on the timing results and the interpretability of them.

  *The final paragraph of the introduction has been updated to better present the novelty of the coherence matrix approach for studying landslide activity*

  *"We also demonstrate that for some landslides, an InSAR coherence matrix approach can be used not only to constrain the timing of new landslides, but also to detect multi-stage failure such as reactivations (i.e. complete failure on one date followed by further failure within or connected to the landslide at a later date) and precursory motion (i.e. displacement on one date followed by complete failure of the same area at a later date)."*

  *At lines 39-44 of the original manuscript has been changed in the revised manuscript to:*

  *"We use the amplitude-based method of Burrows et al. (2022) to constrain the failure timing of new landslides. We also explore an approach based on interferometric SAR (InSAR) coherence matrices, a technique that has successfully been applied in landcover mapping (Giffard-Roisin et al. 2022; Jacob et al. 2020), but not yet tested for landslide timing. Here, we identify landslides where this method appears to identify multi-stage failure such as reactivations (i.e. complete failure on one date followed by further failure within or connected to the landslide at a later date) and precursory motion (i.e. displacement on one date followed by complete failure of the same area at a later date)."*

- Given that the coherence matrix approach is a novel methodology for landslide timing detection, I believe this requires comprehensive analysis on the ability to use it for this purpose. This currently seems to be lacking. For example, once you identify timing, you seem to be 100% sure about the validity. Partly from these timings you then derive conclusions on precursory motion and reactivation. There seem to be little discussion about the actual uncertainties related to those timings and the constraints it puts on the results. What is the effect of noise? In addition, the coherence product has a relatively large spatial resolution. It is unclear how this spatial resolution affects the ability to use this product in your method. I can imagine that mixed pixels might play an important role. Given that you propose a new method, perhaps somehow a sensitivity analysis on the effect of landslide size on the ability

to detect changes could be beneficial, especially when others want to use a similar technique.

Large landslides are more likely to be assigned a timing using InSAR coherence. We have carried out an analysis of this and added information on the sensitivity of the InSAR coherence methods to Section 3.2. Thank you for suggesting that we add this information *"Within the landslides examined, larger landslides were more likely to be assigned a timing by the InSAR coherence methods than smaller landslides. The inventory of Ferrario contained 87 landslides > 10000 m2, 38 in the range 8000-10000 m2, 75 in the range 6000-8000 and 171 in the range 3960-6000 m2 of which 70 (80%), 25 (66%), 46 (61%)  and 86 (50%) were assigned a timing respectively."*

- You use a very limited amount of landslides compared to the complete inventory and derive general conclusions on the triggering conditions of new and reactivated landslides. Are these results representative? I think this should to be addressed and put into perspective.

Yes, large landslides are over-represented in these data. This has been added to Section 3.2 (see response to previous comment for text change). A reference to this has been added to the discussion section at line 301 of the original manuscript as follows:

 *"This observation primarily applies to large landslides, since these are more likely to be assigned a timing by both the amplitude (Burrows et al. 2022) and coherence (Sect. 3.2) methods."*

And in the conclusion

*"We have demonstrated that when a coherence matrix approach is used, we can detect not only single failures but also reactivations and thus build a more complete picture of landslide activity, although such methods cannot be applied to all landslides."*

Has been changed to *"We have assessed a new method for landslide timing detection based on InSAR coherence matrices. This approach, which is mainly applicable to larger events, can detect not only single failures but also reactivations and thus build a more complete picture of landslide activity."*

- I believe that there are some structural changes that could improve the manuscript: (1) I think it is more relevant to first describe the SAR datasets (current section 2.3) before diving into the detection methodologies (current section 2.2). This will allow to introduce all the concepts that you will be talking about during the detection methodologies section. (2) Section 4.1 seems to consist of a mix of methodology and results that I think would better fit in the methods and results sections.

(1) As you suggest, we have exchanged the order of these two sections

(2) Lines 281-294 of the original manuscript have been moved to the methods section. However, we have kept the rest of Section 4.1 in the Discussion rather than moving it to the results as this is not an additional result, but instead builds on the results of the paper.

*"Using the information on landslide evolution through time derived from SAR, we were able to consider the conditions under which new landslides and reactivations were triggered (Section 4.1). For this analysis, estimates of PGV experienced during each earthquake were obtained from the USGS Shakemap webpage (USGS, 2018a, b, c, d). For 19/08, we took the maximum PGV experienced by each landslide during the Mw 5.8, 6.3 and 6.9 earthquakes. In*

*the majority of cases, this was the PGV of the Mw 6.9 earthquake. Slope was calculated from the 30 m Copernicus digital elevation model in Google Earth Engine and the maximum value was taken within each landslide polygon.*

*The landslide probability under these conditions can be estimated with the logistic regression model of Nowicki Jessee et al. (2018) using regression coefficients derived in that study for a global database of landslides. For lithology, we used the coefficient derived for intermediate volcanics, which comprise the majority of the study area according to the global lithological map of Hartmann and Moosdorf (2012) and for landcover, closed deciduous forest, which is the landcover type shared by most of the landslides (Dossa et al., 2013). Although lithology and landcover also affect landslide susceptibility, we do not attempt to control for these: lithology does not vary much across the study area, particularly since many new landslides and reactivations occur on the same scars and so at the same locations. Differences in landcover between landslides is too difficult to account for since the landslides themselves mean that it changes through time."* Has been added as section 2.5 in the revised manuscript.

- It is not very clear which precursory motions and reactivations have been validated through optical data. This seems essential for the applicability of the methods and the interpretability of the results later.

For reactivations, this information is in the final column of table 1. For precursory movements, we could only assess this in one or two cases due to the lack of optical imagery after the first earthquake. It is for this reason, that we suggest that more research is needed. Figure 3 of the revised manuscript shows an additional example of a landslide reactivation that has been detected in coherence matrix (Part of this figure was in figure 2 of the original manuscript)

- Figures should be improved, legends, scale and axis-labels are sometimes missing,

We have made the improvements to the figures that you suggest throughout this review. We have also divided Figure 2 of the original manuscript into two separate figures that show (i) the spatial correlation between the signal seen in the coherence and landslides (in Fig. 2 of the revised manuscript and (ii) example of landslide reactivation seen in both multi-spectral imagery and InSAR coherence

Specific comments Abstract:

❖ To me it is not clear what methodology you have used. Is it a new methodology? And if so, did it work accurately? I think you could be more accurate in that.

To make clearer the methodology used and to specify its novelty, we have made the following change

*"Sentinel-1 techniques"* has been changed to *"Sentinel-1 amplitude and a new coherence-based method"* at line 2. Although the accuracy of the methods is explored in the paper, we do not include it in the introduction.

❖ I think you can be more transparent in how representative the results are, especially considering that you only analyze a fraction of all landslides within the inventory. Are these results representative for the whole event?

It is true that the methods only work for a subset of landslides and this was not made clear in the previous version of the abstract. To address this, we have added the text in bold to the following sentence. *"Overall, we demonstrate that, **although they are not sensitive to all landslides,** Sentinel-1 amplitude and coherence are valuable tools to study how landslide hazard and mass wasting evolve during sequences of triggers."*

Specific comments Introduction

❖ I believe that a more elaborated introduction on the SAR methodologies and the novelty of your work in regards to that would be relevant (around lines 36-39). Also, in the next line you mention InSAR coherence, but this has not been introduced yet (line 40).

*We also demonstrate that for some landslides, an InSAR coherence matrix approach can be used not only to constrain the timing of new landslides, but also to detect multi-stage failure such as reactivations (i.e. complete failure on one date followed by further failure within or connected to the landslide at a later date) and precursory motion (i.e. displacement on one date followed by complete failure of the same area at a later date).*

Changed to:

*We use the amplitude-based method of Burrows et al. (2022) to constrain the failure timing of new landslides. We also explore an approach based on interferometric SAR (InSAR) coherence matrices, a technique that has successfully been applied in landcover mapping (Giffard-Roisin et al. 2022; Jacob et al. 2020), but not yet tested for landslide timing. Here, we identify landslides where this method appears to identify multi-stage failure such as reactivations (i.e. complete failure on one date followed by further failure within or connected to the landslide at a later date) and precursory motion (i.e. displacement on one date followed by complete failure of the same area at a later date).*

❖ From line 39 onwards it starts to read like a conclusion, while I think a presentation of aim and objectives is more suitable. Is the aim to create a methodology to identify precursory and reactivation? Or, are you going to use old methodologies and identify their suitability in identifying that? Or, is the point more process-based and you want to understand the occurrence of landslides as a result of this earthquake sequence?

The text from line 39-44 is included so that the reader will know what they can expect to find in the paper. However, we believe the change made in response to your previous comment better highlights the novelty of the coherence matrix approach, which has not previously been used for landslide timing.

Specific comments Data and Methods

❖ To me it seems more relevant to discuss SAR data and processing (section 2.3) before the SAR detection methodologies (section 2.2). I think it is better to first properly introduce the SAR products and their properties, and how landsliding influence this signal before mentioning the detection methodologies. You have mentioned amplitude and coherence before, but what exactly consist of is unclear until section 2.3. I think a better structure and separation is relevant.

We have reordered Sections 2.2 and 2.3 as you suggest

❖ Figure 1 could use some adjustments to increase readability:

o Inlay of location with respect to larger area needed.

o Fig. 1b could be improved visually. Needs x-axis + label. I find the forward and backslash a bit strange. Perhaps you can use a straight line with different colors, or patterns? In addition, I think a better distinction for the division in time between the two inventories for example by using different background colors (same as you use in fig1 a) would be relevant.

o Perhaps nice and informative to add an elevation map in Fig. 1a

Thank you for these suggestions. We have made the recommended alterations to this figure and amended the caption.

❖ The reason for only taking landslides > 2.000m2 during amplitude analysis and >3.600m2 during coherence analysis is not well explained. Why 2.000m2 where amplitude resolution is 20x22 m (440m2) and 3.600m2 where coherence is 60x66m (3.960m2). This means that coherence resolution is lower than some landslide events?

For the amplitude methods, it is necessary to have multiple pixels within the landslide polygon to calculate the metrics (e.g. pixel variability) used by Burrows et al. (2022). Therefore, the polygon needs to be larger than 440 m2. An analysis of the effect of landslide size on the sensitivity of the method was carried out in that paper.

Text changed at line 101 of the original manuscript from *"These metrics, particularly those relating to geometric shadows and bright spots, work best in forested areas and can be applied to medium-large landslides (> 2000 m2)."* to *"These metrics, particularly those relating to geometric shadows and bright spots work best in forested areas. The method is limited to landslides > 2000 m2 so that each polygon contains enough pixels to calculate metrics (e.g. pixel variability) and is more sensitive for larger landslides (Burrows et al. 2022)."*

For the coherence, you are right that 3960 would be a more logical cutoff. In fact, 3960 was the cutoff that was used in the analysis and 3600 is a mistake in the written manuscript which will be altered in the revised version at lines 73,191,239, 257 of the original draft and in the caption of Figure 3

❖ Following previous point: Particularly for coherence you have a rather low resolution. How does this influence the detection results? Does this mean that some landslides are only covered by one pixel? How does that work with mixed pixels? This could be the case for many landslides right? How does this influence your methodology? This is not really mentioned. In addition to that, I believe that presenting the size distribution of your inventory is important for the interpretation of the results and their accuracy.

The resolution (i.e. the smallest resolvable object) of a coherence map is coarser than the pixel size of the coherence map because of the moving window used in the coherence calculation. Large landslides are more likely to be assigned a timing using InSAR coherence. Information on how the sensitivity of the InSAR coherence methods varies with respect to landslide size has been added to Section 3.2

*"Within the landslides examined, larger landslides were more likely to be assigned a timing by the InSAR coherence methods than smaller landslides. The inventory of Ferrario contained 87 landslides > 10000 m2, 38 in the range 8000-10000 m2, 75 in the range 6000-8000 and 171 in the range 3960-6000 m2 of which 70 (80\%), 25 (66\%), 46 (61\%)  and 86 (50\%) were assigned a timing respectively."*

❖ I'm also curious why you use a low coherence resolution. A moving window not necessarily reduces the resolution of coherence?

The smallest resolvable cell of a coherence map is coarser than that of the SAR image because the moving window used to calculate coherence has a blurring effect. Therefore the moving window reduces the resolution of the coherence compared to the interferogram or to amplitude images.

❖ Since you are proposing a new application of a methodology using the coherence I think this requires some sort of a sensitivity analysis. Given that the land cover is more or less the same, it would be interesting to see the accuracy in regards to size.

Large landslides are more likely to be assigned a timing using InSAR coherence. Information on the sensitivity of the InSAR coherence methods has been added to Section 3.2. Thank you for suggesting this

*"Within the landslides examined, larger landslides were more likely to be assigned a timing by the InSAR coherence methods than smaller landslides. The inventory of Ferrario contained 87 landslides > 10000 m2, 38 in the range 8000-10000 m2, 75 in the range 6000-8000 and 171 in the range 3960-6000 m2 of which 70 (80%), 25 (66%), 46 (61%) and 86 (50%) were assigned a timing respectively."*

❖ In the end you use less than 10% (after size threshold) of the complete inventory. I think this will affect the interpretability of the results, but this is not really mentioned specifically.

The change we have made in response to your previous comment details the percentage of landslides that are timed by InSAR coherence and how this varies according to landslide area.

We have also added the following text at line 301 of the original manuscript to make clear that our interpretation is based on larger events

*"This observation primarily applies to large landslides, since these are more likely to be assigned a timing by both the amplitude (Burrows et al. 2022) and coherence (Sect. 3.2) methods."*

And changed the following text in the conclusions:

*"We have demonstrated that when a coherence matrix approach is used, we can detect not only single failures but also reactivations and thus build a more complete picture of landslide activity, although such methods cannot be applied to all landslides."*

Changed to *"We have assessed a new method for landslide timing detection based on InSAR coherence matrices. This approach, which is mainly applicable to larger events, can detect not only single failures but also reactivations and thus build a more complete picture of landslide activity."*

❖ How it is currently written, I do not fully agree with your reasoning to not use optical data (line 96) in the methodology. Cloud cover does not necessarily have to cover the full event completely. There could always be some cloud free spots, especially in regards to earthquake-triggered landslides (where rainfall doesn't play a major factor). This information can be used and does not necessarily rule out the use of optical.

The landslides triggered by this event have already been studied using optical satellite imagery and the aim here was to complement these pre-existing analyses, particularly that of Ferrario (2019) using InSAR coherence and SAR amplitude. However we have made the following change to make this clearer:

*"Due to prevalent cloud cover in our study area and the fact that the landslides are already somewhat constrained in time since the earthquake timings are known a-priori, we did not expect using optical imagery to offer an advantage here, so we used the method presented in Burrows et al. (2022). This method uses time series of four metrics"* at line 95 of the original manuscript changed to *"The 2018 Lombok, Indonesia earthquake sequence has previously been studied using optical satellite imagery, and it was found that cloud cover during the sequence presented a significant limitation, particularly in differentiating between landslides triggered during the first two earthquakes (Ferrario, 2019). Therefore, here we use the SAR-amplitude method of Burrows et al. 2022, which uses time series of four metrics"*

❖ Rainfall could also play a role in influencing the amplitude and coherence values. Was there any (heavy) rain during the sequence that could have influenced the results?

Not during the earthquake sequence itself. The earthquake sequence occurred during Indonesia's dry season and with little recorded rainfall in August (Ferrario, 2019)

*"This highlights the fact that there are some events for which coherence analysis may be inherently unsuitable"* at line 452 of the original manuscript has been changed to *"The earthquake sequence occurred during the dry season in Indonesia, with little rainfall recorded during the month of August (Ferrario, 2019). However, this sensitivity to soil moisture changes means that there are some events for which coherence analysis may be inherently unsuitable,"* in the revised manuscript.

❖ For figure 2. It seems like the coherence of element (13,14) that indicates post-event stable conditions and element (9,10) that indicates reactivation are almost similar to each other? What does that mean?

This similarity in absolute coherence value for individual coherence maps is why we use the full coherence matrix rather than individual time series. While there is some variation in coherence that is not caused by landslides, the fact that coherence is lower for all the interferograms that span a given earthquake is what indicates that the landslide failed at that time.

New text at line 141 of the revised manuscript *"This allows us to better differentiate between coherence loss due to earthquake-induced landslide activity and coherence loss due to other factors, such as acquisition geometry."*

❖ What polarization are you using and why? I think this needs some elaboration.

We are using vertically polarised SAR images. You are right, this information was missing from the manuscript. We have added it at line 181 of the original manuscript

*"Vertically polarised (VV) imagery was used, since these data are sensitive to land-cover changes in vegetated areas and have been widely used in coherence- and amplitude-based landslide detection methods (e.g. Burrows et al. 2022; Deijns et al. 2022)."*

Specific comments Results

❖ To me it is rather unclear if (and for which landslides) the SAR-based precursory or reactivation conclusions are validated using optical imagery. Now it reads like this is not really the case. Can this difference in timing be explained by inaccuracies of the methodology? Potential noise? Basically, how reliable are these results? I would like to see a figure where it becomes clear the reactivation or precursory movement defined by SAR products is in fact actual precursory movement or a reactivation. From table 1 to me this doesn't become really clear.

Examples of the correspondence between the matrix and preliminary failure/ reactivations are given in figures 3 (new) and 6 (previously 5) of the revised document. Hopefully this will better support our interpretation.

❖ You mention that you are only able to derive only a portion of the landslides. Why is this? I think this is relevant needed information.

The landslide timing is only kept if the signal is strong enough. Landslides without a strong enough signal could be in areas of the interferogram with low background coherence (making the coherence loss due to the landslide relatively small) or in areas highly sensitive to geometric decorrelation (introducing noise). This was described at lines 175-176 of the original manuscript, but we have added the text in bold to make this clearer

*"We chose a minimum threshold of 1.5 standard deviations in order to maximise the accuracy of the method. Raising this threshold beyond 1.5 reduced the number of timed landslides without improving the accuracy (Fig. A2).* **For this reason, we do not obtain timing information for all landslides".**

❖ I have to say that I was a bit confused by the mentioning of all these landslide timing detection numbers throughout sections 3.1-3.3. A visual would probably help with the interpretation. Maybe a better back and forth with figure 3 could benefit this?

Referring to Figure 3 during Sections 3.1 and 3.2 is not possible, since this figure shows the landslide timing information we can obtain by combining amplitude and coherence (i.e. section 3.3). We believe the numbers are necessary to report the results.

❖ As a suggestive question: In the end I wonder about the use of SAR if there is only ~30% (of the 10% of the total inventory) that can be detected. Were you able to identify more accurately using optical (even if it is manual of course)? Perhaps there is many landslides that now have a more accurate timing using optical than SAR? Mentioning this could increase transparency in regards to the applicability of the methods used.

In some cases it may be possible to time more with optical, but the advantage of the SAR is that we have consistency in time (i.e. every landslide that we are able to time is constrained to a 6-day window). For example, here, we would struggle to differentiate between the 3ʳᵈ and 4ᵗʰ earthquakes in the sequence because the imagery between these two is especially cloudy. Timed landslides are also likely to be clustered in space due to gaps in cloud cover and less visible in unvegetated areas. Additionally, the SAR methods are likely to be more useful in the future when more SAR satellites (e.g. NISAR, ROSE-L) with regular acquisition strategies have been launched.

Given the analysis on the impact of landslide size that we have added based on you other comments, we suggest that SAR amplitude is useful for medium-large landslides (>2000), while InSAR coherence is more useful for large landslides.

New text at line 411 of original manuscript *"However, there are limitations, particularly in terms of landslide size, that impact the applicability of the methods. Altogether, we obtained timings for a relatively limited portion of the landslide inventory of Ferrario, 2019. The number of landslides that it is possible to constrain the timing of may improve in the future by incorporating data from planned SAR satellites with regular acquisition strategies such as NiSAR and ROSE-L missions (Jones et al. 2021, Davidson et al. 2021)."*

❖ You seem to assume that when the timing is done, the estimation is 100% correct. The reasoning for this is not clearly explained, what is the uncertainty in this? For example, in line 248 you mention

with certainty that they have to be reactivations. Is this really true? No noise at all? Can it just be that the timing detection is wrong? Especially considering that the coherence doesn't always work correctly as you mention 252-253. I think this is an important aspect to address.

It is difficult to know exactly the accuracy of the InSAR coherence methods since the optical is an imperfect record of landslide reactivation and precursory movements. When the InSAR coherence and optical disagree, this may be due to the InSAR coherence detecting something that is not visible in the optical. This is discussed in Section 4.4, but we have amended this part of the results section and point the reader to this section in the revised version.

*"Possible explanations of the disagreement between the optical and InSAR coherence results are discussed further in Sect. 4.4."* at line 253 of the original manuscript changed to *"If the optical data is assumed to be correct, the accuracy of the InSAR coherence methods thus appears to be 72-80%. However some cases where the optical and SAR disagree may be due to differences in what the two datasets are sensitive to. This is explored further in Sect. 4.4."* in the revised manuscript.

❖ Fig 3:

o Fig. 3f to improve readability maybe increase the size of the circles proportionally

The size of panel f has been increased accordingly

o Fig. 3f, should 112 be 113? Then 258 adds up to 371 as mentioned in line 261

Yes that is correct, thank you for catching this mistake. We have changed 112 to 113 in the figure caption

o I think you should add a reference to the PGA data of the USGS

A reference to USGS shakemap has been added to the figure caption

Specific comments Discussion

❖ The first two paragraphs under line 278-293 would probably better fit in the method section rather than discussion. In addition, the rest of section 4.1 would better fit in the results section.

Lines 281-294 of the original manuscript have been moved to the methods section. However, we have kept the rest of Section 4.1 in the Discussion rather than moving it to the results as this is not an additional result, but instead builds on the results of the paper.

*"Using the information on landslide evolution through time derived from SAR, we were able to consider the conditions under which new landslides and reactivations were triggered (Section 4.1). For this analysis, estimates of PGV experienced during each earthquake were obtained from the USGS Shakemap webpage (USGS, 2018a, b, c, d). For 19/08, we took the maximum PGV experienced by each landslide during the Mw 5.8, 6.3 and 6.9 earthquakes. In the majority of cases, this was the PGV of the Mw 6.9 earthquake. Slope was calculated from the 30 m Copernicus digital elevation model in Google Earth Engine and the maximum value was taken within each landslide polygon.*

*The landslide probability under these conditions can be estimated with the logistic regression model of Nowicki Jessee et al. (2018) using regression coefficients derived in that study for a global database of landslides. For lithology, we used the coefficient derived for intermediate volcanics, which comprise the majority of the study area according to the global lithological map of Hartmann and Moosdorf (2012) and for landcover, closed deciduous forest, which is the landcover type shared by most of the landslides (Dossa et al., 2013). Although lithology and landcover also affect landslide*

*susceptibility, we do not attempt to control for these: lithology does not vary much across the study area, particularly since many new landslides and reactivations occur on the same scars and so at the same locations. Differences in landcover between landslides is too difficult to account for since the landslides themselves mean that it changes through time."* Added as section 2.5 in the revised manuscript.

❖ Fig 4:

o Fig4c legend needed, what do the colors represent?

The colours represent the 4 earthquakes as in panels (a) and (b) with darker tones indicating a higher prevalence of the slope and PGV in the 2D histogram

o Fig4c why not subdividing them into new and reactivations as well?

These are not histograms of landslides but histograms of the whole study area. Thus new and reactivation cannot be differentiated. This is clarified in the figure caption and the axes have been made more readable.

o the yellow color is not properly readable

This colour has been darkened in the revised version to improve visibility

The following changes have been made to figure 4 (figure 5 in the revised document):

*"2D histograms of Slope and PGV across the study area during each earthquake."* changed to *"2D histograms of Slope and PGV during each earthquake. Darker tones indicate a higher prevalence across the study area"* in the figure caption.

Yellow colour has been darkened to improve readability

Axis labels in panel c have had font size increased to improve readability.

❖ I think the representativeness of the results should be discussed. You results do not include smaller landslides and they only consist of a fraction of all the events.

To address this comment, we have added new text at line 411 of original manuscript *"However, there are limitations, particularly in terms of landslide size, that impact the applicability of the methods. Altogether, we obtained timings for a relatively limited portion of the landslide inventory of Ferrario, 2019. The number of landslides that it is possible to constrain the timing of may improve in the future by incorporating data from planned SAR satellites with regular acquisition strategies such as NiSAR and ROSE-L missions (Jones et al. 2021, Davidson et al. 2021)."*

❖ Figure 5:

o Adding a scale bar seems essential for the interpretation in relation with coherence resolution.

o No x-label and y-label for 5e, and no x-label for the legend

Changes made to figure 5 (fig. 6 in revised document)

- Scale bar added to panels a-d
- Label added to legend of 5e
- X and y axes labelled as "Time" as in Figure 2

❖ In line with previous comments, I'm not entirely convinced that this coherence change indicates precursor movement, my main concerns being:

o How does the resolution affect these processes, especially given that you have quite a low coherence resolution? Also, how does the mixing signal in pixels influence this?

This landslide polygon is large in size, a scale bar has been added to the figure to better show this

o Isn't there any uncertainty in the coherence based timing? The uncertainty is low since the timings have to be tied to an earthquake. The accuracy of the methods is already discussed in the results section.

o Wouldn't you need a longer time series, to derive a trend (like Dini et al., 2022 and Jacquemart & Tiampo, 2021)? Perhaps it is just a noisy image? Maybe there is still some effect of local clouds, or local rainfall?

It is true we cannot say that it is definitely precursory movement, that is why the section is titled "**Possible** detection of precursory motion during the 28/07 earthquake". However, it is unlikely to have been caused by rainfall since there was very little rainfall during the month of August. It is also unlikely to be just a noisy image since the coherence loss is not seen for every landslide polygon.

Using a longer time series is not feasible since we are not seeing a slow acceleration to failure as in those studies but two periods of movement (in eqs 1 and 2). The movement in eq1 is considered precursory because it is not visible in the optical yet.

Changes made:

- Information on rainfall during the earthquake sequence has been added to the revised manuscript according to your earlier comment.
- *"This highlights the fact that there are some events for which coherence analysis may be inherently unsuitable"* at line 452 of the original manuscript has been changed to *"The earthquake sequence occurred during the dry season in Indonesia, with little rainfall recorded during the month of August (Ferrario, 2019). However, this sensitivity to soil moisture changes means that there are some events for which coherence analysis may be inherently unsuitable,"* in the revised manuscript.

o Figure 5:

▪ This figure would greatly benefit from adding a coherence map. Then we can see the coherence response, and even see how the coherence pixels cover the landslide location. Now it is not transparent.

A time series of coherence map has instead been added to Figure 2 to show how coherence pixels cover the landslide location. Thank you for suggesting this.

▪ From images 18-06-2018 to 01-08-2018 you can see that some of the grasses have been removed. Can the coherence loss be attributed to this vegetation change?

The process that resulted in the coherence loss needs to have taken less than 6 days. Overall, we do not expect vegetation growth or dieback to occur this quickly. Therefore, processes relating to the earthquake are more likely to have resulted in the coherence loss. Agricultural practices (e.g., grass mowing) cannot be ruled out as the source of coherence loss, but they are expected to i) affect a single pair of images and ii) be located where agricultural fields are present. The use of the full

coherence matrix assures that a longer time span is considered. Finally, we underline that the majority of the study area is covered by forest, where vegetation changes due to agriculture are less significant.

▪ If the precursory movement is due to the 28/07, why is the coherence rather low the image before this event as well (element (5,6))? Or, am I interpreting it incorrectly?

This element is fairly low, but others (e.g. 4,6) are not, suggesting this was not a permanent change but was caused by geometric decorrelation/ small difference in atmosphere or soil moisture. This is the reason that we use the full coherence matrix as opposed to the time series of 6-day interferograms alone

❖ NISAR will have a different wavelength than Sentinel-1. This can impose differences and alter the usability of your methodology. This might require elaboration.

We have added new text in revised manuscript to discuss this point:

*"The number of landslides that it is possible to constrain the timing of may improve in the future by incorporating data from planned SAR satellites with regular acquisition strategies such as NiSAR and ROSE-L missions (Jones et al., 2021a; Davidson and Furnell, 2021). The longer wavelength of these satellites is likely to improve their landslide detection capacity in forested areas as they will undergo less decorrelation caused by the movement of vegetation (Burrows et al., 2020). However further testing will be needed to establish this."*

❖ How has this perpendicular baseline effect affected your results? Could it have induced noise?

The main effect is to reduce the number of landslides for which coherence analysis was possible, not to give wrong timings. Landslides for which the coherence matrix is too noisy are not assigned a timing. We have clarified this by added the text in bold to the text at line 175-176 of the revised manuscript.

*"We chose a minimum threshold of 1.5 standard deviations in order to maximise the accuracy of the method. Raising this threshold beyond 1.5 reduced the number of timed landslides without improving the accuracy (Fig. A2).* **For this reason, we do not obtain timing information for all landslides".**

❖ Were there any rainfall events in your study area during the earthquakes that have affected the results?

Not during the earthquake sequence itself. The earthquake sequence occurred during Indonesia's dry season and with little recorded rainfall in August (Ferrario, 2019)

*"This highlights the fact that there are some events for which coherence analysis may be inherently unsuitable"* at line 452 of the original manuscript has been changed to *"The earthquake sequence occurred during the dry season in Indonesia, with little rainfall recorded during the month of August Ferrario (2019). However, this sensitivity to soil moisture changes means that there are some events for which coherence analysis may be inherently unsuitable,"* in the revised manuscript.

❖ Figure 6:

o Meaning of the color? Legend required. Which image pairs do these dots indicate?

The legend to this figure has been added in the revised version of the manuscript. The coloured dots show the coherence of pre-event (blue), co-event (orange), post-event (indigo) and unknown (grey) interferograms

❖ Figure 7:

o When did this event occur? Why is there little more consistent higher coherence post-event?

Based on the matrix, this landslide was active in all earthquakes. The low coherence in the post-event pixels is due to geometric decorrelation – these two interferograms had a long perpendicular baseline.

o There is little explanation on Fig. 7b. I think it could use some more elaboration.

New text added at line 451 of the original manuscript "The rainfall event can be seen by plotting the absolute difference in rainfall in the three days before each image used to form an interferogram were acquired (Fig. 8b)." (figure number has changed due to new figure 3 added to revised manuscript)

o X- and y-labels needed for both subplots

X and y labels have been added to the plot

Main comments Conclusion

❖ You mention that this study combines optical and SAR. However, this is not what you mention in the introduction and methods. You use optical as validation, right?

We did use the optical for validation, but here we refer to the fact that the landslides were mapped using optical imagery in the original study of Ferrario (2019) and then SAR data were used to gain information on their evolution in time.

*"This study represents one of the first combined applications of optical imagery and Sentinel-1 amplitude and coherence to depict the multi-stage failure following a sequence of earthquakes."* At line 474 of the original manuscript changed to *"This study represents one of the first combined applications of optical imagery and Sentinel-1 amplitude and coherence to study landslide multi-stage failure following a sequence of earthquakes"* in the revised manuscript.

Line-by-line comments

Line 3: Adding a timeframe in which the sequence occurred here would be relevant

This information has been added to the revised manuscript: *"during the 2018 Lombok, Indonesia earthquake sequence"* has been changed to *"during an earthquake sequence that occurred over a 23-day period in 2018 in Lombok, Indonesia."*

Line 5/6: Here you use 'many' a lot, I think you should be more accurate and present some percentages.

Around half of the landslides for which we derived timings from coherence were active in more than one earthquake in the sequence.

*"While the majority of new landslides were triggered during the largest earthquake in the sequence on 05/08, we are also able to identify landslide activity associated with other, lower magnitude earthquakes on 28/07, 09/08 and 19/08, with many landslides active in more than one earthquake."* Changed to *"While the majority of new landslides were triggered during the largest earthquake in the sequence on 05/08, we are also able to identify landslide activity associated with other, lower magnitude earthquakes on 28/07, 09/08 and 19/08, with around half of the landslides studied active in more than one earthquake."*

Line 8: I find this sentence slightly unclear 'weakening effect' of what? Could probably benefit from some rephrasing

The weakening effect refers to the fact that the presence of landslide scars after an earthquake means there is more unconsolidated material that can then be remobilised by later earthquakes. This is elaborated on further later in the manuscript, we do not feel it is necessary to include it in the abstract.

Line 13-14: I find the meaning of 'significant mass wasting effect' to be a bit unclear

*"significant mass wasting effect"* has been changed to *"source of erosion"* at line 13 of the revised manuscript.

Line 15: Possible addition: 'In particular, earthquake-induced landslide inventories' to be more precise in which type of landslide inventory you are addressing.

*"landslide inventories"* has been changed to *"earthquake-triggered landslide inventories"* as you suggest.

Line 24: What do you mean by: 'cumulative effect'?

This has been removed in response to your next comment.

Line 24-27: I find this sentence to be a bit unclear. What is the point of this sentence? I would advise to rephrase for clarity.

*"The cumulative effect of such earthquake sequences on rapid, shallow landsliding is difficult to study as it requires satellite images to be acquired between each earthquake, but aftershock-triggered landslides can represent a considerable part of the total landslides for some events (Ferrario, 2019; Tanyas et al., 2022)."* At lines 24-27 has been changed to *"The evolution of triggered landslides during such earthquake sequences is difficult to study as it requires satellite images to be acquired between each earthquake, but aftershock-triggered landslides can represent a considerable part of the total landslides for some events (Ferrario, 2019; Tanyas et al., 2022)."* In the revised manuscript

Line 31: The references here are not ordered the same as the others, maybe better to add the relevant reference after each earthquake?

The references here are ordered according to the listed earthquakes

Line 32-33: The point you mention about medium resolution not being able to capture reactivation or remobilization is not that clear. It now reads like SAR will be able to provide a solution, but how will SAR be able to fix this while the resolution is lower?

The spatial resolution of SAR is lower but the wavelength is a few cm and the scale of things detectable by SAR is determined by this, not only by the pixel size.

*"Satellite synthetic aperture radar (SAR) data may offer a solution to this problem as these data can be acquired through cloud cover and are sensitive to landslides."* At line 34 of the original manuscript changed to *"Satellite synthetic aperture radar (SAR) data may offer a solution to this problem as these data can be acquired through cloud cover and are sensitive to cm-scale movements at the Earth's surface including landslides."* At line 34-35 of the revised manuscript

Line 60-61: 'few landslides were triggered by this earthquake' . I'm wondering where this statement is based on if there is no imagery available. I think it will be helpful to clarify that. It's based on the cited references and on the limited cloud-free portions of the imagery that was available.

Line 68: Does these number total to ~15.000 or are the inventories overlapping? It is slightly unclear. The inventories are overlapping, we have revised the text to clarify this.

*"Ferrario (2019) mapped 4823 landslides (with a total area of 4.88 km2) following the 05/08 earthquake and 9319 (10.25 km2) at the end of the sequence (Fig 1a)."*

Changed to

*"Ferrario (2019) mapped 4823 landslides (with a total area of 4.88 km2) following the 05/08 earthquake increasing to 9319 (10.25 km2) following the 19/08 earthquakes (Fig 1a)."*

Line 72-73: It is not fully clear if these 991 and 371 landslides are only due to this size threshold, or if there were other factors on which the reduction is based?

We only applied a size threshold. The text has been rewritten to clarify this.

*"For this reason, we limit the amplitude analysis in this study to 991 landslides > 2000 m2 (following Burrows et al., 2022) and the coherence analysis to 371 landslides > 3600 m2 (the size of the coherence window in Sect. 2.3)."* at line 72 of the original manuscript changed to *"For this reason, we limit the amplitude analysis to landslides > 2000 m2 (following Burrows et al. 2022, 991 events) and the coherence analysis to landslides > 3960 m2 (the size of the coherence window in Sect. 2.3, 371 events)."*

Equation 2: Slightly unclear what it does, what is i, what is n?

i is each pixel, n is the number of pixels in the boxcar (i.e. the number used in the summation). "i" has been italicised in the revised manuscript in the description of this equation.

Fig. 2: I'm wondering if it might be more useful to add the actual dates to get a better understanding of the temporal baseline as well?

We have not added the dates as it would make the Figure too busy. Since the time interval is consistent throughout (6 days) it is not necessary. This is stated at line 188 of the original manuscript

*"Sentinel-1 collected images every six days on two tracks throughout the earthquake sequence (Fig. 1b)."*

Since we have restructured by swapping the order of section 2.2 and 2.3 according to your earlier comment, this will now be written before the Figure is printed. Hopefully this should make things clearer.

Line 75: Differential InSAR comes a bit out of the blue and maybe should be introduced first?

This paragraph has been deleted following the comments from Reviewer 2.

Line 82-84: Wouldn't this sentence be more appropriate at the end of the introduction? Here you identify that it still has an exploratory aspect.

This paragraph has been deleted following the comments from Reviewer 2.

Line 88-90: I think soil moisture should be added here as well, since it influences the amplitude values.

This line states that "The amplitude of the signal returned to the satellite depends on the scattering properties of the material that this energy interacts with at the Earth's surface."

Soil moisture is something that determines the scattering properties along with roughness etc. Therefore, "scattering properties" includes soil moisture already and we do not think it is necessary to change the text here.

Line 91: This seems to contradicts your point in line 81 where you mention that there are multiple methodologies for this.

There are multiple methodologies using either SAR amplitude *or* InSAR coherence. Only 2 use SAR amplitude, so line 91 does not contradict line 81.

Line 103: 'very little prior knowledge on timing': Although this is a relative statement, I would argue that you have rather a lot of information already, a few months accuracy. I would instead mention this few months accuracy instead of 'very little'.

*"where very little prior knowledge on landslide timing would be available."* At line 103 of the original manuscript has been changed to *"where landslide timing can usually only be constrained to within a few months"*

Line 104: 'are concurrent with one of the earthquakes': I'm not an expert in earthquake related landslides, but are there no landslides that occur a few days after the shock? There is no doubt whatsoever?

The vast majority of landslides will occur simultaneously with the earthquake (at least when we are using a 6-day temporal resolution to study them. It is possible that a landslide could occur a few days after the shock through progressive failure, particularly if it rained in the days after an earthquake. But the vast majority of landslides will be concurrent with the earthquake, so the assumption is valid.

Line 148: I think it is a bit unclear how this full matrix approach might be able to do that. I think some elaboration is required.

*"Finally, since previous studies have shown that coherence is sensitive not only to the denudation of the hillslope that can be captured by the amplitude method described in Sect. 2.2.1, but also to precursory movements and to movement of material in unvegetated areas, the full matrix approach might be able to reveal multiple failure stages."*

Changed to *"Finally, previous studies have shown that coherence is sensitive not only to the denudation of the hillslope that can be captured by the amplitude method described in Sect. 2.2.1, but also to precursory movements and to movement of material in unvegetated areas. Thus, coherence might be able to reveal multiple failure stages, with the matrix approach providing a more reliable indicator of landslide activity than pairwise coherence time series"*

Line 155-160: Why is there a difference between the coherence values before and after the earthquake? You use typically and generally, but it is unclear why. For better interpretability I would relate these values to the actual landscape conditions.

The difference is caused by the change in landcover – the bare rock exposed by the landslide has a higher coherence than the pre-event vegetation did.

*"Since both were acquired after the earthquake sequence had ended, and thus after the landslide had occurred, coherence is high."* At line 155 of the original manuscript changed to *"Since both were*

*acquired after the earthquake sequence had ended, and thus after the landslide had denuded the hillslope, coherence is high."*

Line 163: 'has failed more than once': I'm curious how certain this is? Can it just be noise?

To support this statement, we have divided Figure 2 in the original manuscript into two so that we can show evidence of multi-stage failure in multi-spectral imagery alongside the matrix. We have then added the following text at line 169 of the original manuscript

*"This is supported by multi-spectral satellite imagery acquired over this landslide during the earthquake sequence, in which we first see the loss of vegetation within the landslide scar following the 05/08 earthquake (Fig. 3c) and then see the extent of this denuded area grow following the 19/08 earthquake (Fig. 3d)."*

Line 170-176: I have to admit, that this part is a bit unclear to me. I think this section would benefit some from elaborating a bit more on 'our analysis' and how this relates to equation 1.

To make this clearer, we have rewritten lines 170-171 in the original manuscript

*"To make best use of this information, we carried out our analysis in two separate stages: first with the pre-event and co-event image pairs to identify the first failure and then with the co-event and post-event image pairs to identify the final failure."*

has been rewritten as *"To allow for detection of multi-stage landslide failure, we carried out our analysis in two separate stages: first identifying the first failure timing that minimises the residuals (Eq. 1) when dividing the pre-event and co-event image pairs and then repeating this with the co-event and post-event image pairs to identify the final failure timing."*

Line 195: This first sentences seems unnecessary

*"In Sect. 3, we present the landslide timing results obtained from the SAR amplitude and coherence methods described in Sect. 2.2. In order to validate these results, we compare with the timing information that can be obtained from optical and multi-spectral images acquired during the earthquake sequence."* (lines 195-197 of original manuscript) has been rewritten as

*"In order to validate the landslide timing information derived from SAR, we compare with the timing information that can be obtained from optical and multi-spectral images acquired during the earthquake sequence."* Deleting the first sentence

Line 201-203: This use of optical data contradicts your initial statement that optical data is not relevant for timing detection here. I think that requires some rephrasing

*"In some cases, we were then able to further constrain the timing using cloud-free areas of multi-spectral Sentinel-2 and Planet images and high-resolution optical images in Google Earth Explorer"*

At lines 201-202 has been changed to *"In areas that were cloud-free in Planet, Sentinel-2 or Google Earth images acquired between the 28/07 and 05/08 or the 09/08 and 19/08, we were able to carry out a more precise validation."* To make this clearer

Line 204: 'Second, many landslides fail more than once': How do you know this? This has not really been clearly presented in the inventory section.

*"This change in total area includes landslides polygons mapped on 05/08 that grew in size in the final inventory, indicating landslides that failed more than once during the sequence."* Has been added to the inventory section at line 69 of the original manuscript.

Line 211: 'fitted' what does this mean? Little unclear

We mean the shape of the landslide visible in Google Earth or Sentinel-2 is better delineated by the 05/08 polygon. To clarify this, *"fitted by"* has been changed to *"delineated by"*

Line 227: 'represent the main failure' Why is that? I think this could use some elaboration

This was described at line 219. We have made the following change to clarify that at line 227 (and elsewhere) we use "main" failure to refer to the largest change in area of the landslide polygon.

*"SAR timings derived from amplitude (Sect. 2.3.1), which primarily detect denudation of the hillslope were assessed against the timing of the largest failure by area in the optical images."* At line 219 of the original manuscript changed to *"SAR timings derived from amplitude (Sect. 2.3.1), which primarily detect denudation of the hillslope were assessed against the timing of the largest failure by area in the optical images (referred to as the "main" failure)."*

Table 1: you use 05/07 instead of 05/08 for optical timing

Thank you for identifying this mistake, this has been corrected in the revised manuscript.

Line 239-244: While you present the same type of results, you do it differently (with/without percentages/ different way of telling it). I think it is better readable if there is a similar structure.

*"Of these, 19 initiated during the earthquake on 28/07, 40 on 05/08, none on 09/08 and 2 on 19/08."* Has been changed to *"This was the 28/07 earthquake in 19 cases (31%), 05/08 in 40 cases (66%) and 19/08 in 2 cases (3%)."* To better match the following sentence and improve readability as you suggest.

Line 246: At first, this number (153 of the 213) confused me a little bit since you first say 61 and then 213, but I see now that you mean difference between optical and SAR. I think this should be clarified.

Yes, we were comparing optical and SAR

*"Overall, the two timings agree for 153 of the 213 landslides (72%, Table 1)"*

At line 246 of the original manuscript has been changed to *"Overall, the final failure timing agrees with the optical imagery for 153 of the 213 landslides (72%, Table 1)"* in the revised manuscript to make this clearer

Line: 258: '214' should be 213, right?

Yes that is correct, this has been corrected in the revised manuscript

Line 258: '170' where does this value come from? In sect 3.1 you mention 307. Or, is this only for the >3600 m2 landslides?

Yes 170 refers only to the landslides > 3960 that we have done the coherence analysis on.

*"From the amplitude methods, we have timing information for 170 landslides"* at line 258 of the original manuscript has been changed to

*"From the amplitude methods, we have timing information for 170 landslides > 3960 m2"*

Line 260: First mention of the figure 3 is Fig 3.f shouldn't it be Fig. 3.a then?

The labelling of the panels is designated by their location in the figure, we cannot change panel f to panel a without reorganising the figure.

Line 265-266: I think this statement requires backup in the form of results or supplementary material for transparency.

To support this statement, we have added Figure 3 to the revised manuscript and added a reference to it here.

Line 272: Is it 259 or 258 (as in line 261).

This should have been 258, we have corrected it. Thak you for noticing this mistake.

Line 296: '2.5cm/s' Even lower right? 2.1/2/2 Yes correct, 2.1 would be a better value since the lowest is 2.07. We have corrected this in the revised manuscript

Line 299: degree sign needs to be changed

The degree sign has been changed in the revised manuscript

Line 345-346: References should be between brackets? Ending with a full stop

This has been corrected in the revised manuscript

Line 350-351: I think this statement should be backed-up by some reasoning.

To better back up this statement, we have added the text in bold in the revised manuscript

*"Overall, we believe that the earthquakes on 09/08 and 19/08 resulted in more landslide activity than they would have done had they not been part of the sequence, **since activity associated with these earthquakes occurred at low PGV (Fig. 4b)**"*

Line 359-360: This sentence seems a bit unclear? Don't you mean to say that the spatial extent is already defined by the landslides that occurred in the main shock? Maybe some rephrasing is needed.

*"However, since this activity takes the form of reactivations rather than new failures, its spatial extent is controlled by the shaking intensity experienced in the mainshock"* at line 359-360 of the original manuscript has been changed to *"However, since this activity takes the form of reactivations rather than new failures, its spatial extent is determined by the locations of triggered landslides, and thus shaking intensity associated with the mainshock"* in the revised manuscript

Line 362 & 367: What do you mean by 'mass wasting effect'? amount of regolith mobilized?

Yes we mean the total volume eroded by the landslides. *"Mass wasting effect"* has been changed to *"erosional effect"* to make this clearer

Line 381-384: You say it is 'particularly likely' but why is that?

We have rephrased this sentence to remove "particularly likely"

*"This is particularly likely for the 6 landslides that were not visible until 05/08 in the optical imagery, but were detecting as failing on both 28/07 and 05/08 by the coherence matrix; and for the 4 landslides that were mapped in the second half of the sequence by Ferrario (2019), but were active in every earthquake according to the coherence matrix."* Has been changed to *"These 12 landslides*

*include 6 that were not visible until 05/08 in the optical imagery, but were detecting as failing on both 28/07 and 05/08 by the coherence matrix; and 4 that were mapped in the second half of the sequence by Ferrario (2019), but were active in every earthquake according to the coherence matrix."*

Line 388: Instead of reactivations, could noise be a factor?

The optical data alone is not sufficient to assess whether these are noise rather than reactivations. We have added the following text at line 390 of the original manuscript to discuss this:

*"To confirm that these are indeed reactivations, comparison against a different dataset, such as field surveys, ground-based SAR or high resolution DEMs would be necessary, but this is beyond the scope of this study."*

Line 444: I think this should be included as a reference, right? We have added this as a reference rather than a url in the revised manuscript

---

## Referee Report (RR1)

**Dear,**

Thanks for implementing my comments in the manuscript. The manuscript shows clear improvements and seems nearly ready for publication. I have just a few minor additional comments that I believe could further strengthen the text in some parts.

**Abstract:**

• In my opinion, I would like to see some clarity on the fact that the methods work on larger landslides just like you have in the conclusion that will directly imply that the results are based on larger landslides as well

Line 12: You have to mention the uncertainty for the precursor in line with your discussion. Use word like 'potential' or 'possible'

Line 14: I assume you mean the sensitivity related to size. Therefore I think you should be more explicit. This can be rephrased to something like: 'Overall, we demonstrate that, although they are more sensitive to larger landslides, Sentinel-1 amplitude and coherence'.. etc

**Introduction:**

Line 38: I would add (in bold): 'medium resolution **optical** imagery such as .. 'and potentially remove 'medium resolution' since your reply emphasize the point of difference in what it can capture rather than the spatial resolution.

Line 44: Typo in 'introducing noise in a and coherence'

Line 54: INSAR acronym already defined, not needed again.

Line 55: Brackets needed for references.

**Data and Methods**

I would like to see a one sentence addition that highlights how noise factors are differentiated from a landslide in the coherence matrix. This would make it much clearer. Now you say that it provides a more reliable indicator (line 184-185 and 193-197 but it does not seem fully clear how), even after paragraph in line 195-206. Just a one sentence addition could clarify this I think. This puts everything a bit more in perspective and allows to understand better where potential inaccuracies would come from.

Line 76: Regarding my initial comment. This sentence is a bit confusing. On the one hand there is no possibility to map landslides, but on the other hand you say that they agree that there is a few landslides. That is confusing. I would adapt it to something like: 'cloud cover prevented **from comprehensive** landslide mapping'.

Line 85/86: make sure km2 is in superscript

Line 116 Brackets needed for the references

Line 128: Deijns et al., 2022 also uses amplitude to define the timing of landslides. Although they don't use it to identify the timing of **individual** landslides. Perhaps this is what you mean, but it requires some clarification.

Line 205: I would relate this to landscape since it is likely more vegetated in contrast to the bare soil after the event. This will make it clearer as to why this is visible.

Line 208: I would really like to see explained shortly what differentiates coherence loss due to landslide activity from other noise factors (like mentioned in line 186) in the coherence matrix. That would really ease the understanding of the figure. You explain it in your reply but it is not clear from the text.

**Results**

Table 1: 05-07 should be changed to 05-08.

Line 322: There still seems to be too much confidence from this. Especially given your sentence after that. You should add something like 'likely' or 'potential' reactivations.

Line 333-334: Maybe add comma for thousands. E.g., 10,000

Line 338: Explanation of 'main' failure is not needed anymore since you explained this earlier.

Fig 4. I have to say that 4c reactivation is not that clear. (blue outline on green)

**Discussion**

Fig. 5.: 1% dashed line is lighter in panel a than panel b, this should be homogenized.

Line 461: I would like to see mentioned some examples of these inaccuracies for clarity and interpretability of the method.

Line 472 I would use 'could' or 'would suggest that these 29' etc. to add nuance.

Line 491: Section 4.2 highlights that the precursory motions are quite uncertain. I would be a bit more cautious in phrasing this here. Better to phrase it as 'possible precursory movement, just like you mention in the section 4.2 title.

Line 565: 'Potential' or 'Possible' precursory activity to be in line with your discussion

---

## Author Response (AR2)

Thank you for your time in reviewing our manuscript. Below you will find our responses in green along with changes made in the revised version of the manuscript.

**Abstract:**

In my opinion, I would like to see some clarity on the fact that the methods work on larger landslides just like you have in the conclusion that will directly imply that the results are based on larger landslides as well. We have added this information – see response to your comment on line 14.

Line 12: You have to mention the uncertainty for the precursor in line with your discussion. Use word like 'potential' or 'possible'

We have added the words in bold to the following sentence at line 11 of the revised manuscript: "We also identified an example where **possible** precursory motion **detected** during the first earthquake in the sequence was later followed by larger scale failure."

Line 14: I assume you mean the sensitivity related to size. Therefore, I think you should be more explicit. This can be rephrased to something like: 'Overall, we demonstrate that, although they are more sensitive to larger landslides, Sentinel-1 amplitude and coherence' .. etc

While size is an important factor determining whether landslides are detectable, it is not the only one. Other relevant factors include pre-event landcover type and slope aspect relative to the SAR sensor. However, since size is one of the most important factors, we have added this information:

"Overall, we demonstrate that, although they are not sensitive to all landslides, Sentinel-1 amplitude and coherence..." at line 12 of the revised manuscript changed to

"Overall, we demonstrate that, although they are not sensitive to all landslides and are more likely to detect larger events, Sentinel-1 amplitude and coherence..."

**Introduction:**

Line 38: I would add (in bold): 'medium resolution **optical** imagery such as .. ' and potentially remove 'medium resolution' since your reply emphasize the point of difference in what it can capture rather than the spatial resolution. We have added the word "optical" at line 36 of the revised manuscript. The "medium" we have left in since it is relevant – reactivations can be visible in high resolution images (as in Figure 6).

Line 44: Typo in 'introducing noise in a and coherence' This has been changed to "altering the amplitude of SAR images and introducing noise and decreasing coherence in SAR interferograms." At line 43 of the revised manuscript

Line 54: INSAR acronym already defined, not needed again. Thank you for identifying this mistake. At line 50 of the revised manuscript the acronym definition has been removed.

Line 55: Brackets needed for references. Thank you for identifying this mistake. Brackets have been added in the revised version.

**Data and Methods**

• I would like to see a one sentence addition that highlights how noise factors are differentiated from a landslide in the coherence matrix. This would make it much clearer. Now you say that it provides a more reliable indicator (line 184-185 and 193-197 but it does not seem fully clear how), even after paragraph in line 195-206. Just a one sentence addition could clarify this I think. This puts everything

a bit more in perspective and allows to understand better where potential inaccuracies would come from. At line 168, we have added "All interferograms formed from images spanning the time when the landslide failed will have low coherence, providing a signal that is distinct from other possible causes of noise such as variations in soil moisture and acquisition geometry."

Line 76: Regarding my initial comment. This sentence is a bit confusing. On the one hand there is no possibility to map landslides, but on the other hand you say that they agree that there is a few landslides. That is confusing. I would adapt it to something like: 'cloud cover prevented **from comprehensive** landslide mapping'. In fact, neither study mapped any landslides following this first earthquake. However, we have added the word "comprehensive" at line 71 of the revised manuscript as you suggest since it would have been possible in some parts of the study area.

Line 85/86: make sure km2 is in superscript. We have corrected this at line 80/81.

Line 116 Brackets needed for the references. We have corrected this at lines 99 and 103

Line 128: Deijns et al., 2022 also uses amplitude to define the timing of landslides. Although they don't use it to identify the timing of **individual** landslides. Perhaps this is what you mean, but it requires some clarification. We have changed "shallow landslides" to "individual shallow landslides" at line 114 of the revised manuscript. You are correct, Deijns et al. are timing whole inventories of landslides rather than individual events

Line 205: I would relate this to landscape since it is likely more vegetated in contrast to the bare soil after the event. This will make it clearer as to why this is visible. You are correct, thank you for the suggestion. At line 178 of the revised manuscript, we have made the following change:

"Figure 2a shows an example of a coherence matrix for a landslide in the Lombok study area that failed during the 05/08 earthquake" changed to

"Figure 2a shows an example of a coherence matrix for a landslide in a forested part of the study area that failed during the 05/08 earthquake"

Line 208: I would really like to see explained shortly what differentiates coherence loss due to landslide activity from other noise factors (like mentioned in line 186) in the coherence matrix. That would really ease the understanding of the figure. You explain it in your reply but it is not clear from the text. At line 188 of the revised manuscript we have added the following explanation: "Other factors that affect coherence such as changes in soil moisture and acquisition geometry are more variable in time and as such do not result in distinct patches of high and low coherence in the matrix."

**Results**

Table 1: 05-07 should be changed to 05-08. Thank you for identifying this error, this has been changed in the table.

Line 322: There still seems to be too much confidence from this. Especially given your sentence after that. You should add something like **'likely'** or **'potential'** reactivations. At line 311 of the revised manuscript, we have changed "Reactivations" to "These probable reactivations".

Line 333-334: Maybe add comma for thousands. E.g., 10,000 We have added a comma for numbers 10,000 or higher

Line 338: Explanation of 'main' failure is not needed anymore since you explained this earlier. While it is true that this was explained earlier, we find it clearer to reiterate it here to help the reader

understand how we combine the amplitude-derived "main" failure with the "first" and "last" failures derived from coherence.

Fig 4. I have to say that 4c reactivation is not that clear. (blue outline on green) We have changed to a darker shade of green so that the two colours are more distinct in the revised manuscript.

**Discussion**

Fig. 5.: 1% dashed line is lighter in panel a than panel b, this should be homogenized. Thank you for identifying this, it has been corrected in the revised version.

Line 461: I would like to see mentioned some examples of these inaccuracies for clarity and interpretability of the method.

"While some of these may simply be due to inaccuracies in the SAR methods or manual landslide mapping, there are patterns that suggest that some of them may be explained by differences in what is and is not detectable in SAR and optical satellite imagery" has been changed to

"Inaccuracies in the SAR methods or manual landslide mapping, such as features being mapped as landslides that are in fact something else, may account for some of these disagreements. However, there are patterns that suggest that some of them may instead be due to differences in what is and is not detectable in SAR and optical satellite imagery." at line 409-410 of the revised manuscript

Line 472 I would use 'could' or 'would suggest that these 29' etc. to add nuance. We have made the following change at line 421.

"This suggests that these 29 last detected failures were reactivations rather than new failures." Changed to "This suggests that these 29 last detected failures could be reactivations rather than new failures."

Line 491: Section 4.2 highlights that the precursory motions are quite uncertain. I would be a bit more cautious in phrasing this here. Better to phrase it as 'possible precursory movement, just like you mention in the section 4.2 title. We have added "possible" at line 440.

Line 565: 'Potential' or 'Possible' precursory activity to be in line with your discussion. We have added the word "possible" at line 511 and "potential" at line 512.